# JAXLEY: differentiable simulation enables large-scale training of detailed biophysical models of neural dynamics

**Michael Deistler** [1,2] ✉, **Kyra L. Kadhim** [2,3], **Matthijs Pals** [1,2], **Jonas Beck** [2,3], **Ziwei Huang**[2,3], **Manuel Gloeckler**[1,2], **Janne K. Lappalainen** [1,2], **Cornelius Schröder** [1,2], **Philipp Berens** [2,3], **Pedro J. Gonçalves** [1,2,4,5,6] & **Jakob H. Macke** [1,2,3,7] ✉

Biophysical neuron models provide insights into cellular mechanisms underlying neural computations. A central challenge has been to identify parameters of detailed biophysical models such that they match physiological measurements or perform computational tasks. Here we describe a framework for simulating biophysical models in neuroscience—JAXLEY—which addresses this challenge. By making use of automatic differentiation and GPU acceleration, JAXLEY enables optimizing large-scale biophysical models with gradient descent. JAXLEY can learn biophysical neuron models to match voltage or two-photon calcium recordings, sometimes orders of magnitude more efficiently than previous methods. JAXLEY also makes it possible to train biophysical neuron models to perform computational tasks. We train a recurrent neural network to perform working memory tasks, and a network of morphologically detailed neurons with 100,000 parameters to solve a computer vision task. JAXLEY improves the ability to build large-scale data- or task-constrained biophysical models, creating opportunities for investigating the mechanisms underlying neural computations across multiple scales.

Computational models are used to devise hypotheses about neural systems and to design experiments to investigate them. When building such models, a central question is how much detail they should include: Models of neural systems range from simple rate-based point neuron models to morphologically detailed biophysical neuron models. The latter provide fine-grained mechanistic explanations of cellular processes underlying neural activity, typically described as systems of ordinary differential equations[1–3].

However, it has been highly challenging for neuroscientists to create biophysical models that can explain physiological measurements[3–5] or that can perform computational tasks[6,7]. It is hardly ever possible to directly measure all relevant properties of the system with sufficient precision to constrain all parameters directly, necessitating the use of inference or fitting approaches to optimize free model parameters. However, finding the right parameters for even a single-neuron model with only a few parameters can be difficult[5,8], and large-scale morphologically detailed biophysical network models may have thousands of free parameters governing the behavior of ion channels (for example, maximal conductance), synapses (for example, synaptic conductance or time constant) or neural morphologies (for example, radius or branch length).

Recently, in many domains of science such as particle physics, geoscience and quantum chemistry, differentiable, GPU-accelerated

[1]Machine Learning in Science, University of Tübingen, Tübingen, Germany. [2]Tübingen AI Center, Tübingen, Germany. [3]Hertie Institute for AI in Brain Health, University of Tübingen, Tübingen, Germany. [4]VIB-Neuroelectronics Research Flanders (NERF), Leuven, Belgium. [5]IMEC, Leuven, Belgium. [6]Departments of Computer Science and Electrical Engineering, KU Leuven, Leuven, Belgium. [7]Department Empirical Inference, Max Planck Institute for Intelligent Systems, Tübingen, Germany. ✉e-mail: michael.deistler@uni-tuebingen.de; jakob.macke@uni-tuebingen.de

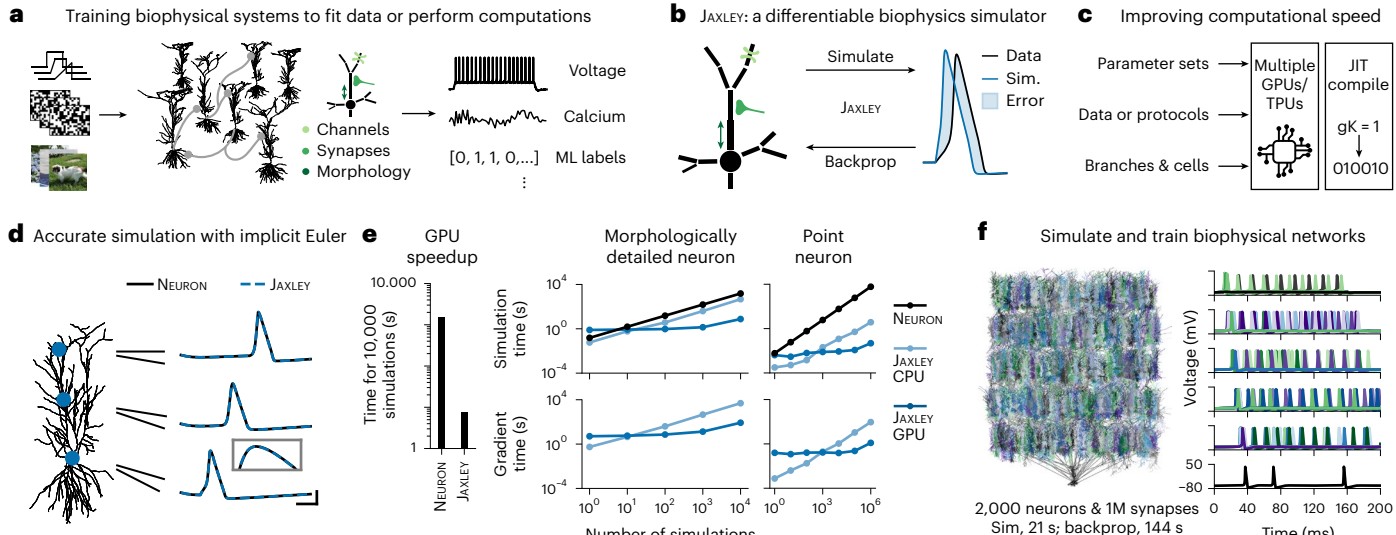

**Fig. 1 | Differentiable simulation enables training biophysical neuron models. a**, Schematic of goal: training biophysically detailed neural systems. **b**, Schematic of method: our simulator, Jaxley, can simulate biophysically detailed neural systems, and it can also perform backprop. **c**, Jaxley can parallelize simulations on (multiple) GPUs/TPUs, and it can just-in-time (JIT) compile code. **d**, Reconstruction of a CA1 neuron[24] and responses to a step current obtained with the Neuron simulator and with Jaxley. Inset is a zoom-in view of the peak of the action potential. Scale bars, 3 ms and 30 mV. **e**, Left: time to run 10,000 simulations with Neuron on a CPU and with Jaxley on a GPU. Right: Simulation time (top) for the CA1 neuron shown in **d** and for a point neuron, as a function of the number of simulations. Bottom: same as top, for computing the gradient with backprop. **f**, Biophysically detailed network built from reconstructions of CA1 neurons (left) and its neural activity in response to step currents to the first layer (right). Runtimes were evaluated on an A100 GPU. M, million; ML, machine learning; Sim., simulation.

simulators have enabled parameter inference for even complicated models using automatic differentiation techniques[9–11]. Such differentiable simulators make it possible to train simulators with gradient descent methods from deep learning: Backpropagation of error ('backprop') makes the computational cost of computing the gradient of the model with respect to the parameters independent of the number of parameters, making it possible to efficiently fit large models. In addition, GPU acceleration allows computing the gradient for many inputs (or model configurations) in parallel, which allows fitting simulations to large datasets.

Numerical solvers for biophysical models in neuroscience are used extensively, and several software packages exist, in particular the commonly used Neuron simulation environment[12]. Yet, none of these simulators allows performing backprop, and currently used simulation engines are primarily CPU-based, with GPU functionality only added post hoc[13–15]. As a consequence, state-of-the-art methods for parameter estimation in biophysical neuron models are based on gradient-free approaches such as genetic algorithms[8,16] or simulation-based inference[17], which do not scale to models with many parameters.

Inspired by the capabilities of deep learning to adjust millions (or even billions) of parameters given large datasets, we here propose to optimize biophysical parameters with gradient descent. To this end, we developed Jaxley, a toolbox for biophysical simulation which, unlike previous simulation toolboxes for biophysical models, can compute the gradient with backprop. In addition, Jaxley leverages GPU acceleration to speed up training. We apply Jaxley to a series of tasks, ranging from fitting physiological data (that is, match experimental recordings such as voltage or calcium measurements[8,18]) to solving computational tasks[7,19] (Fig. 1a). We show that gradient descent can be orders of magnitude more efficient than gradient-free methods, and that it enables training biophysical networks with 100,000 parameters. This unlocks possibilities for data-driven and large-scale biophysical simulations in neuroscience.

## Results
### Jaxley is a differentiable simulator for neuroscience
Jaxley is a Python toolbox for simulation and training of biophysical neuron models. Jaxley implements numerical routines required for

efficiently simulating biophysically detailed neural systems, so-called implicit Euler solvers, in the deep learning Python framework JAX[20]. The automatic differentiation capabilities of JAX enable Jaxley to use backprop to efficiently compute the gradient with respect to any biophysical parameter, including ion channel, synaptic or morphological parameters (Fig. 1b).

For computational speed, Jaxley implements differential equations such that networks, parameter sets or input stimuli can be processed in parallel on GPUs, providing speedups for datasets (via stimulus parallelization) or for parameter sweeps (via parameter parallelization)[13,15]. Jaxley further speeds up simulation and training with just-in-time compilation (Fig. 1c).

Training biophysical models with gradient descent leads to instabilities resulting from parameters having different scales, networks having a large computation graph[21] and loss surfaces being non-convex. Jaxley implements methods that have been developed to overcome these specific issues in deep neural networks (Extended Data Fig. 1). For example, it implements parameter transformations, multilevel checkpointing[22] and optimizers for non-convex loss surfaces (for example, Polyak gradient descent[23]). Furthermore, we designed Jaxley with a user-friendly interface, allowing neuroscientists to build biophysical models (for example, for inserting recordings, stimuli and channels into various branches or cells, or implementing different connectivity structures such as sparse or dense connectivity) and to use automatic differentiation and GPU parallelization. In a dedicated library open to the community, it also implements a growing set of ion channel and synapse models. Jaxley is fully written in Python, which will make it easy for the community to use and to add functionality to it. Jaxley is openly available at https://github.com/jaxleyverse/jaxley/.

### Jaxley is accurate, fast and scalable
We benchmarked the accuracy, speed and scalability of Jaxley for simulation of biophysical models. First, we evaluated the accuracy of Jaxley by creating biophysically detailed multicompartment models of a CA1 pyramidal cell in the rat hippocampus[24,25] and of four layer 5 neurons in the mouse visual area from the Allen Cell Types Database[26]. Every model contained sodium, potassium and leak channels in all branches.

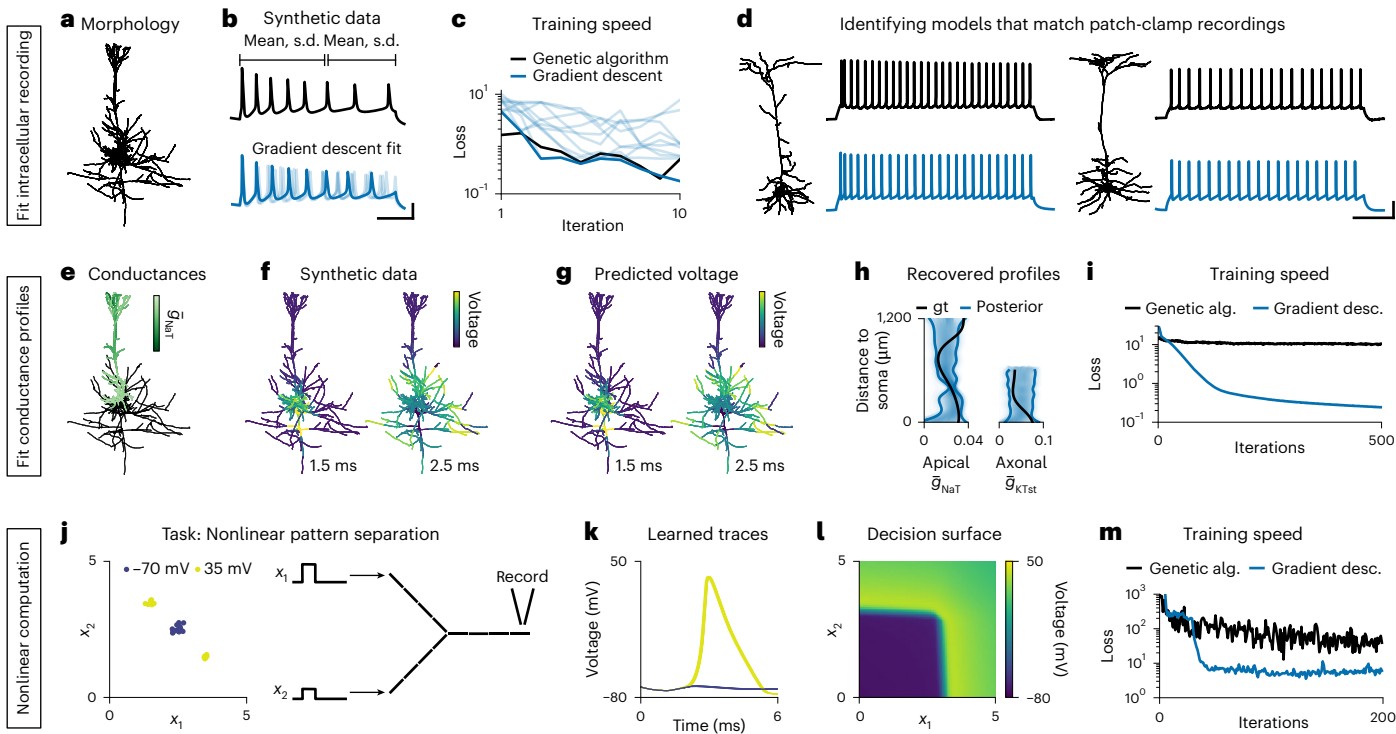

**Fig. 2 | Inferring single-neuron models with gradient descent. a**, (Task 1) Morphology of an L5PC. **b**, Top: synthetic somatic voltage recording (black) and windows that are used to compute summary statistics (top). Bottom: fits obtained with gradient descent. Best fit in dark blue; fits from independent runs in light blue. Scale bars, 20 ms and 30 mV. **c**, Loss value of individual gradient descent runs (light blue), their minimum (dark blue), in comparison to the minimum loss across ten genetic algorithm runs (black). **d**, Neuron morphologies and patch-clamp recordings (black) in response to step currents from the Allen Cell Types Database. Gradient descent fit in blue. Additional models in Extended Data Fig. 5. Scale bars, 200 ms and 30 mV. **e**, (Task 2) Synthetic conductance

profile of an L5PC morphology. **f**, Simulated voltages given the synthetic conductance profile after 1.5 ms and 2.5 ms. **g**, Predicted voltages of gradient descent fit. **h**, Ground-truth (gt) conductance profile as a function of distance from the soma (black) and 90% confidence interval obtained with multi-chain gradient-based Hamiltonian Monte Carlo. **i**, Loss for gradient descent and genetic algorithm. **j**, (Task 3) Nonlinearly separable input stimulus amplitudes (left), and a simplified morphology with 12 compartments (right). **k**, Voltage traces of model found with gradient descent. **l**, Decision surface of the model reveals nonlinear single-neuron computation. **m**, Minimum loss across ten independent runs for gradient descent and genetic algorithm. alg., algorithm; desc., descent.

We stimulated the soma and recorded the voltage at three locations across the dendritic tree. Jaxley matched the voltages of the Neuron simulator at sub-millisecond and sub-millivolt resolution (Fig. 1d and Extended Data Fig. 2).

Next, we evaluated the simulation speed of Jaxley on CPUs and GPUs. We simulated the above-described CA1 cell and a single-compartment model for 20 ms. On a GPU, Jaxley was much faster for large systems or many parallel simulations, with a speedup of around two orders of magnitude (Fig. 1e and Supplementary Fig. 1). For single-compartment neurons, Jaxley could parallelize the simulation of up to 1 million neurons, thereby allowing fast parameter sweeps. On a CPU, Jaxley was at least as fast as Neuron.

We then evaluated the computational cost of computing the gradient with Jaxley. For backpropagation, the forward pass must be stored in-memory, which can easily correspond to terabytes of data for large neural systems. To overcome this, Jaxley implements multilevel checkpointing[22], which reduces memory usage by strategically saving and recomputing intermediate states of the system of differential equations. We found that depending on the simulation device (CPU/GPU/TPU), the number of simulations, the simulated time and the loss function, computing the gradient was between 3 and 20 times more expensive than running the simulation itself (Fig. 1e).

Finally, we show that in addition to parallelizing across parameters (or across stimuli), Jaxley can parallelize across branches or compartments in a network. We built a network consisting of 2,000 morphologically detailed neurons with Hodgkin–Huxley dynamics, connected by 1 million biophysical synapses (3.92 million differential equation states

in total; Fig. 1f). On a single A100 GPU, Jaxley computed 200 ms (that is, 8,000 steps at $\Delta t = 0.025$ ms) of simulated time in 21 s. We then used backprop to compute the gradient with respect to all membrane and synaptic conductances in this network (3.2 million parameters in total), which took 144 s. Estimating the gradient with finite differences—as would be required for packages that do not support backprop—would take more than 2 years (3.2 million forward passes, 21 s each). With simplified morphologies, Jaxley can scale backprop to networks with many millions of synapses on a single GPU (Extended Data Fig. 3).

## Fitting single-neuron models to intracellular recordings

Having demonstrated the accuracy and speed of Jaxley, we applied it to a series of tasks that demonstrate how Jaxley opens up opportunities for building task-driven or data-driven biophysically detailed neuroscience models in a range of scenarios. As a first proof-of-principle, we applied Jaxley to fit single-neuron models with few parameters. We built a biophysical neuron model based on a reconstruction of a layer 5 pyramidal cell (L5PC) (Fig. 2a). The model had nine different channels in the apical and basal dendrites, the soma and the axon[16], with a total of 19 free parameters. We learned these parameters from a synthetic somatic voltage recording given a somatic step-current stimulus with a known set of ground-truth parameters (Fig. 2b).

We used gradient descent to identify parameter sets that minimize the mean absolute error to summary statistics of the voltage trace. Because gradient descent requires differentiable summary statistics, but commonly used summary statistics of intracellular recordings—such as spike count—can be discrete or non-differentiable,

we used the mean and standard deviation of the voltage in two time windows[17]. Starting from randomly initialized parameters, gradient descent required only nine steps (median across ten runs) to find models whose voltage traces are visually similar to the observation (Fig. 2b). A state-of-the-art indicator-based genetic algorithm (IBEA)[16] required similarly many iterations, although each iteration of the genetic algorithm used ten simulations. As a consequence, gradient descent required almost ten times fewer simulations than the genetic algorithm, and, despite the additional cost of backpropagation, found good parameter sets in less runtime than the genetic algorithm on a CPU (Fig. 2c and Extended Data Fig. 4a).

We then used gradient descent to identify parameters that match patch-clamp recordings of four cells from the Allen Cell Types Database[26]. We inserted the same set of ion channels, but, to account for the diversity and complexity of the experimental recordings, we made six additional parameters trainable and used a loss function based on dynamic time warping (DTW). To lower the fitting time caused by the length of the recordings (1 s versus 100 ms in the synthetic experiments), we initially fitted only the first 200 ms of these traces and then added an additional step to fit the entire trace.

We first used gradient descent with a low computational budget (ten runs with ten iterations each; loss in Extended Data Fig. 4b) and found that the resulting traces roughly matched the firing rate of experimental recordings, but did not yet match other features such as spike frequency adaptation (Extended Data Fig. 5). To improve the fits, we used the ability of JAXLEY to parallelize several fitting runs. We parallelized 1,000 gradient descent runs on a GPU and found parameter sets whose voltage traces closely resembled experimental recordings (Fig. 2d and Extended Data Fig. 5). Using JAXLEY, we also parallelized a genetic algorithm on a GPU and found that the resulting fits were of similar quality (Extended Data Fig. 5). Overall, these results demonstrate the ability of gradient descent to fit biophysical models to intracellular recordings, being competitive with state-of-the-art genetic algorithms even on tasks for which those have been extensively optimized.

### Fitting single-neuron models with many parameters

How does gradient descent scale to models with large numbers of parameters? We demonstrate here that, in contrast to genetic algorithms, gradient descent can optimize a single-neuron model with 1,390 parameters.

We used the above-described model of an L5PC. Unlike in the above experiments, we fit the maximal conductance of ion channels in every branch in the morphology, thereby allowing us to model effects of nonuniform conductance profiles[27]. This increased the number of free parameters to 1,390. To generate a synthetic recording, we assigned a different maximal conductance to each branch (sampled from a Gaussian process), depending on the distance from the soma (Fig. 2e). We recorded the voltage at every branch of the model in response to a 5-ms step-current input (Fig. 2f). Experimentally, such data could be obtained, for example, through voltage imaging.

We used gradient descent to identify parameters that match this recording, with a regularizer that penalizes the difference between parameter values in neighboring branches[27]. Despite the large number of parameters, gradient descent found a parameter set whose voltage response closely matched the observed voltage throughout the dendritic tree (Fig. 2g). To understand how much the whole-cell voltage recording constrains the parameters, we used Bayesian inference (implemented with gradient-based Hamiltonian Monte Carlo) to infer an ensemble of parameter sets all of which match the observed voltage (Extended Data Fig. 6). The resulting ensemble revealed regions along the dendritic tree at which the conductance profile was strongly constrained by the data (for example, the transient sodium channel, Fig. 2h, around 400 μm), but it also revealed conductance profiles that were only weakly constrained by the data (Fig. 2h and Extended Data Fig. 6)[2]. Finally, we compared our method with an indicator-based genetic

algorithm[8] and, as expected, we found that access to gradients leads to better convergence: While gradient descent converged to values of low loss within 100 iterations, the genetic algorithm had two orders of magnitude higher loss even after 500 iterations (Fig. 2i).

### Nonlinear single-neuron computation

Next, we trained a single-neuron model to solve a nonlinear pattern separation task on its dendritic tree. While it has been demonstrated extensively that single-neuron models respond nonlinearly to inputs[28], it has so far been difficult to train biophysically detailed neurons on a particular task. Here, we show that stochastic gradient descent enables training single-neuron models with dendritic nonlinearities to perform nonlinear computations.

We defined a simple morphology consisting of a soma and two dendrites, and inserted sodium, potassium and leak channels into all neurites of the cell. We then learned ion channel densities as well as length, radius and axial resistivity of every compartment (72 parameters in total) for the neuron to have low somatic voltage (−70 mV), when both dendrites were stimulated with step currents of intermediate strength, and high somatic voltage (35 mV) when one of the dendrites was stimulated strongly and the other one weakly (Fig. 2j). Therefore, the two classes were not linearly separable, requiring the neuron to perform a nonlinear computation.

After training the parameters with gradient descent, we found that the cell indeed learned to perform this task and spiked only when one dendrite was stimulated strongly (Fig. 2k), effectively having a nonlinear decision surface (Fig. 2l). We again compared gradient descent to an indicator-based genetic algorithm and found that gradient descent finds regions of lower loss more quickly than genetic algorithms (Fig. 2m).

Overall, these results show that gradient descent performs better than gradient-free methods in models with many parameters, opening up possibilities for studying at scale biophysical mechanisms throughout the full neuronal morphology.

### Hybrid retina model of dendritic calcium measurements

So far, we have learned parameters of single-neuron models using small datasets consisting of few stimulus–response pairs. Many models of neural systems, however, consist of multiple neurons, and datasets can contain thousands of stimulus–response pairs[7,29,30]. Using a network model of the mouse retina, we demonstrate that JAXLEY can simultaneously infer cell-level and network-level parameters, such that model simulations match large-scale datasets.

We consider transient Off alpha retinal ganglion cells (RGCs) in the mouse retina, which show compartmentalized calcium signals in their dendrites in response to visual stimulation[18]. To understand the mechanistic underpinning of this behavior, we built a hybrid model with statistical and mechanistic components: We modeled photoreceptors as a convolution with a Gaussian filter, bipolar cells as point neurons with a nonlinearity and an RGC as a morphologically detailed biophysical neuron, with six different ion channels distributed across its soma and dendritic tree (Fig. 3a). To model the fluorescence signal of the calcium indicator, we convolved the intracellular calcium (from the calcium channel of the model) with a calcium kernel (Fig. 3b).

Using JAXLEY, we trained the hybrid model to predict dendritic calcium on 15,000 pairs of checkerboard noise stimuli and calcium recordings. We learned synaptic conductances from the bipolar cells onto the RGC (287 synaptic parameters), as well as cellular RGC parameters (320 cell parameters). After training, we evaluated the trained model on a held-out test dataset. The model had a positive Pearson correlation coefficient with the experimental recording on 146 of 147 recording sites, with an average correlation of 0.25, and a maximum of 0.51 (Fig. 3c).

Next, we tested whether the trained model also reproduced the compartmentalized structure of calcium responses, which has been

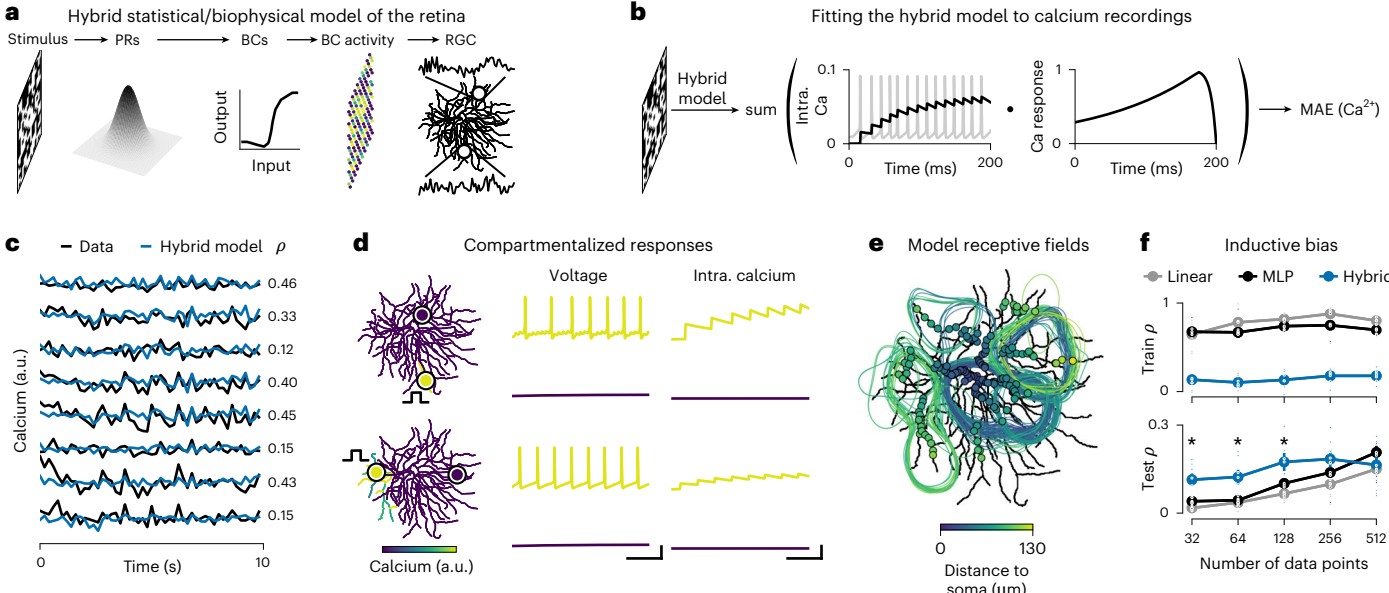

**Fig. 3 | Hybrid model of the calcium responses of an RGC. a**, Schematic of experimental setup and hybrid model. **b**, Schematic of training procedure and loss function. **c**, Measured and model-predicted calcium response across 50 noise images (200 ms each). **d**, Left: calcium response (color map) of the trained hybrid model to a step current to a single branch indicated by step-current sketch. Middle: voltage activity of the model at two branches, one at the stimulus site and one at a distant branch. Right: intracellular calcium concentration in the same two recording sites. Scale bars, 50 ms, 30 mV and 0.025 mM.

**e**, Receptive fields of the hybrid model. **f**, Pearson correlation coefficient between experimental data and model for train (top) and test (bottom) data, for a linear network, a multilayer perceptron, and the hybrid model. Error bars show the s.e.m. over seven datasets (Methods). Asterisks denote a statistically significant difference between mean correlations of hybrid model and multilayer perceptron (MLP; one-sided $t$-test at $P < 0.05$, $P = \{0.0030, 0.0044, 0.017, 0.066, 0.80\}$). a.u., arbitrary units; BCs, bipolar cells; intra., intracellular; MAE, mean absolute error; PRs, photoreceptors.

experimentally measured[18]. We stimulated the model at a distal branch and recorded the model calcium response across all branches of the cell. We found that the calcium signal in response to local stimulation did not propagate through the entire cell (Fig. 3d). In addition, the model receptive fields did not cover the entire cell and were roughly centered around the recording locations (Fig. 3e). This demonstrated a compartmentalized response of the model and qualitatively matched the receptive fields obtained from experimental measurements[18].

The mechanistic components of the model, including the anatomical structure, provide an inductive bias. Therefore, we investigated whether this inductive bias of the hybrid model could lead to better generalization to new data, especially when training data are scarce. We trained a linear model, a two-layer perceptron and the hybrid model on reduced datasets of recordings and compared their performance on a held-out test set (Fig. 3f). While the linear model and the perceptron performed better than the hybrid model on training data, the hybrid model performed better on held-out test data, when little training data were available. These results indicate that the inductive bias brought by the hybrid model effectively can limit the amount of overfitting in the model, suggesting that hybrid components could be used as regularizers for deep neural network models of neural systems[31].

Our results demonstrate that gradient descent enables fitting networks of biophysical neurons to large calcium datasets and allows simultaneous learning of cell-level and network-level parameters.

### Biophysical RNNs solve working memory tasks

To understand how computations are implemented in neural circuits, computational neuroscientists aim to train models to perform tasks[7,19]. In particular, recurrent neural networks (RNNs) have been used to form hypotheses about population dynamics underlying cognition. Typically, such RNNs consist of point neurons with rate-based or simplified spiking dynamics, which prevents studying the contribution

of channel dynamics or cellular processes. We here show how Jaxley makes it possible to train biophysical models of neuronal networks to perform such tasks.

We implemented in Jaxley an RNN consisting of Hodgkin–Huxley-type neurons with a simplified apical and basal dendrite, with each neuron equipped with a variety of voltage-gated ion channels[4]. We sparsely connected the recurrent network with conductance-based synapses and obtained the outputs from passive readout units (Fig. 4a).

We first investigated the dynamics in this biophysical RNN before training. As with rate-based RNNs, these dynamics were strongly dependent on a global scaling factor (called 'gain') of all recurrent synaptic maximal conductances[32]. Our RNN transitioned from a stable to a chaotic regime when the gain was increased, with an intermediate region, where networks displayed regular firing (Fig. 4b). The ability of Jaxley to perform automatic differentiation allowed us to quantify the stability of networks by numerically computing Lyapunov exponents[33] (Fig. 4c and Supplementary Fig. 2).

We then trained the biophysical RNN to perform two working memory tasks, starting with a perceptual decision-making task requiring evidence integration over time[34]. We built a network of 20 recurrent neurons and stimulated each recurrent neuron with a noisy time series with either positive or negative mean value (Fig. 4d). We trained input weights, recurrent weights and readout weights (109 parameters) such that the network learned to differentiate between positive and negative stimuli during a response period after 500 ms. Despite the long time horizon of this task (500 ms, corresponding to 20,000 time steps of the simulation), gradient descent found parameters such that the RNN was able to perform the task (Fig. 4d and Supplementary Fig. 3), with the voltage in the readout neurons differentiating the input means with 99.9% accuracy across 1,000 trials. To solve this evidence integration task, some input, recurrent and output weights were pushed toward zero and the remaining weights were pushed toward their positive and negative constraints during training (Fig. 4e).

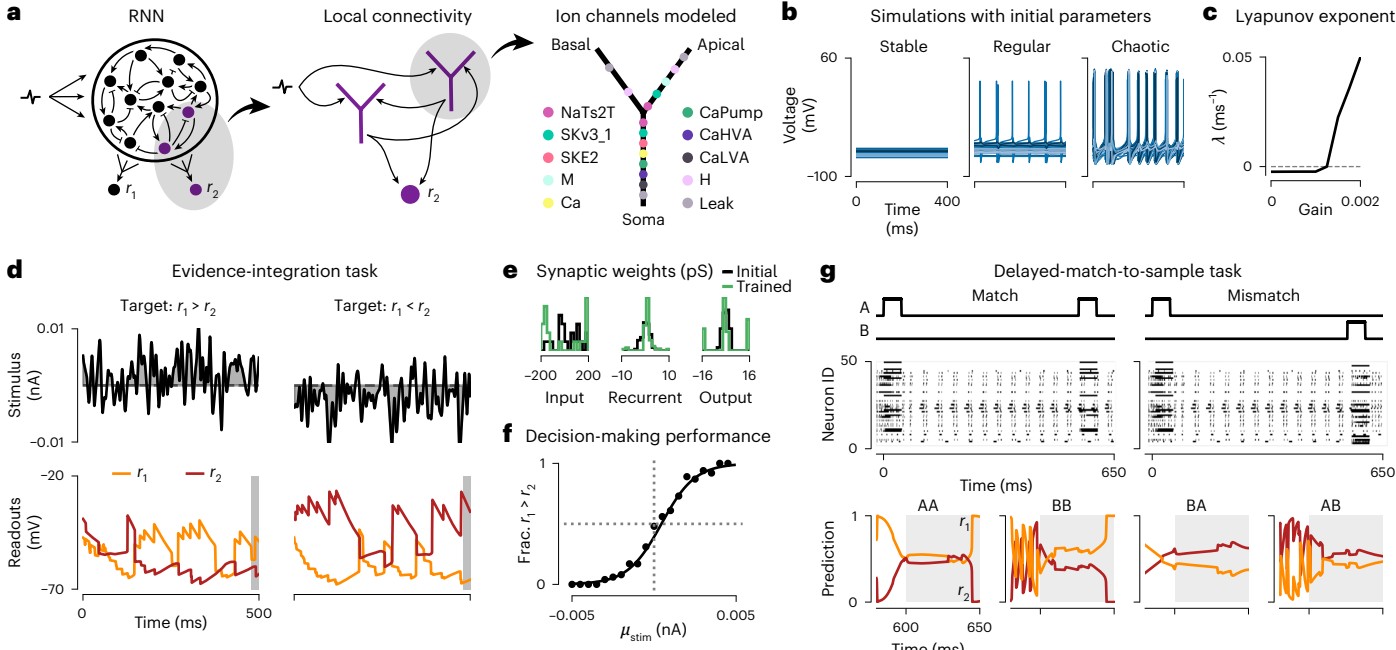

**Fig. 4 | RNN models with Hodgkin–Huxley-type neurons perform working memory tasks. a**, Schematic of the RNN. **b**, Autonomous dynamics of the recurrently connected neurons before learning the parameters for different values of synaptic gain. **c**, Maximal Lyapunov exponent indicates transition to chaos with increasing synaptic gain. **d**, (Task 1) Evidence integration task. Gaussian noise stimulus (top) and voltage traces of the two readout neurons.

Response period in gray. **e**, Histogram of initial and trained input, recurrent and output weights of the network. **f**, Psychometric curve showing the fraction of times the RNN reported the summed input to be greater than zero ($r_1 > r_2$), as a function of the stimulus mean. **g**, (Task 2) Delayed-match-to-sample task. Two stimuli separated by a delay (for two of four input patterns), raster plots of the trained network activity, and readout neuron prediction.

We also evaluated the generalization abilities of the trained biophysical RNN. We varied the mean value of the positive and negative stimuli and found that the relationship between average response and stimulus mean closely resembled the well-known sigmoidal psychometric curve of decision-making, where the network more often failed when the stimulus had a lower signal-to-noise ratio (Fig. 4f). In addition, the trained RNN generalized to evidence integration tasks of longer durations (Extended Data Fig. 7).

We next used the RNN to solve a more challenging working memory task, a delayed-match-to-sample task, where the RNN had to maintain information over an extended period of time[19,35]. We trained the RNN to classify patterns, consisting of two step-current inputs with a delay between them, into matching (same identity of the inputs) or non-matching (different identity of the inputs; Fig. 4g). We used curriculum learning to solve this task. By training input, recurrent and readout weights, as well as synaptic time constants of a network with 50 recurrent neurons (542 parameters), we found parameter sets that solved the task and correctly classified all four patterns (Fig. 4g). We then inspected the population dynamics of the trained biophysical RNN and found that the network used a form of transient coding to solve the task (Extended Data Fig. 8).

Overall, these results demonstrate that gradient descent allows training RNNs with biophysical detail to solve working memory tasks. This will allow a more quantitative investigation of the role of cellular mechanisms contributing to behavioral and cognitive computations.

### Training biophysical networks with 100,000 parameters

Finally, we show that gradient descent enables training of large biophysical models with thousands of cellular-level and network-level parameters on machine learning-scale datasets to solve classical computer vision tasks such as image recognition.

We implemented a feedforward biophysical network model in JAXLEY and trained it to solve the classical MNIST task, without

artificial nonlinearities such as ReLU activations. The network had three layers: The input and output layers consisted of neurons with ball-and-stick morphologies and the hidden layer consisted of 64 morphologically detailed models obtained from reconstructions of CA1 cells (Fig. 5a)[15,24]. The network was interconnected by biophysical synapses. We trained sodium, potassium and leak conductances of every branch in the circuit (55,000 parameters), as well as all synaptic weights (51,000 parameters).

We simulated the network for 10 ms, as this was the time it took for the stimulus to propagate through the network. After training with stochastic gradient descent, upon being stimulated with a '0' digit, the softmax of the voltages of the output neurons indicated a high probability for the digit '0' (Fig. 5b). The network achieved an accuracy of 94.2% on a held-out test dataset, which is higher than a linear classifier, demonstrating that the biophysical network used its nonlinearities to improve classification performance. The biophysical network, however, performed slightly worse than a multilayer perceptron with ReLU nonlinearities, suggesting either that the spike/no-spike nonlinearities are more difficult to train than ReLU nonlinearities, or that the (binary) spike/no-spike representations lead to lower bandwidth than graded ReLU activations (Fig. 5c). After training, the biophysical network also developed interpretable hidden-layer tuning (Extended Data Fig. 9).

How do the learned parameters of the biophysical network contribute to its ability to classify MNIST digits? Surprisingly, we found that the ranges of the trained synaptic parameters were roughly similar to the ranges of the untrained network and that the membrane channel conductances were roughly centered around their initial value (Fig. 5d). This does not mean that the learned values of these parameters do not contribute to the learned network dynamics: While the ranges of parameters did not change substantially, individual parameters could vary largely (Extended Data Fig. 10). Furthermore, resetting subsets of parameters to their initial value reduced classification performance, sometimes to chance-level accuracy (10%; Fig. 5e). This indicates that

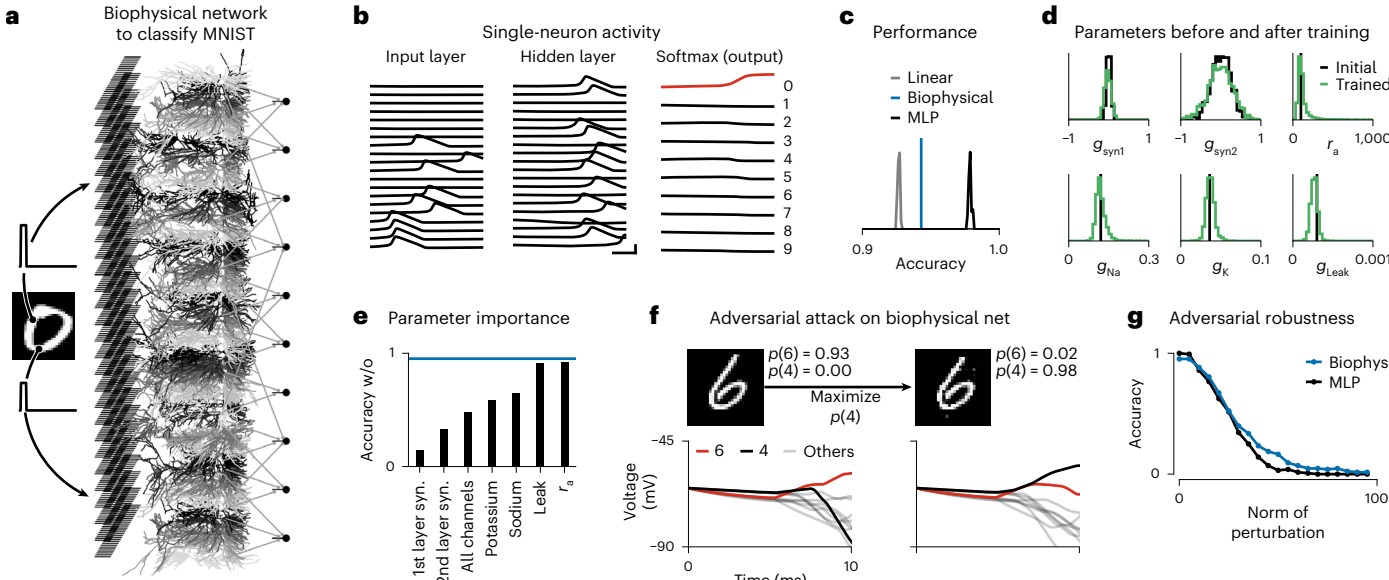

**Fig. 5 | Training biophysically detailed networks to solve computer vision tasks. a**, Biophysical network consisting of 28 × 28 input neurons, 64 morphologically detailed hidden neurons, and 10 output neurons. **b**, Voltages (left and middle) and softmax probabilities of the output neurons (right) measured at the somata of neurons in the trained network in response to an image labeled as '0'. Red color in the output layer indicates the image label. Scale bars, 2 ms and 80 mV. **c**, Histograms of test-set accuracy of 50 linear networks (gray), 50 multilayer perceptrons with 64 hidden neurons and ReLU activations (black) and the biophysical network (blue). **d**, Histogram of parameters before (black) and after (green) training. $g_{syn1}$, $g_{syn2}$, synaptic conductances in the first and second layer, respectively. **e**, Test-set accuracy for trained network when subsets of parameters are reset to their initial value. Blue line indicates the fully trained network. **f**, Clean image (top, left) and adversarial image perturbation (top, right), as well as corresponding voltage traces of the output neurons (bottom). **g**, Accuracy across 128 test-set examples, as a function of the norm of the adversarial image perturbation. Biophys., biophysical.

biophysical simulations built purely from the aggregate statistics of measured parameter values could not be sufficient for the models to develop the ability to perform tasks[4].

Finally, efficient access to gradients not only allows training biophysical models to match physiological data or perform a task, but also opens up possibilities to perform new kinds of analyses, such as evaluating adversarial robustness. We stimulated the biophysical network with an image showing a '6', and then altered minimally the image with gradient descent such that the network would classify it as a '4' (Fig. 5f). While the biophysical network classified the initial image as a '6' with high confidence, the perturbed image was classified as a '4'. We compared the adversarial robustness of the biophysical network to a trained multilayer perceptron. We found that the two networks are similarly vulnerable to adversarial attacks (Fig. 5g), in contrast to previous studies suggesting that biophysical networks could largely improve adversarial robustness[15].

Overall, these results demonstrate that the ability of JAXLEY to perform backprop in biophysical models can be used to train large-scale networks at full biophysical detail. In particular, backprop overcomes virtually any computational limit on the number of parameters of biophysical networks that can be included in the optimization.

## Discussion

Backprop and computational frameworks that provide efficient, scalable and easy-to-use implementations have been key to advances in deep learning. They have made it possible to efficiently optimize even very big systems with gradient descent. However, for biophysical neuron models, currently used tools[12,36–38] cannot perform automatic differentiation (comparison to other toolboxes in Supplementary Note 1). We here presented JAXLEY, a computational framework for differentiable simulation of neuroscience models with biophysical detail. JAXLEY can perform automatic differentiation through its differential equation solver, thereby enabling backprop to compute the gradient with respect to virtually any biophysical parameter.

We expect that JAXLEY will enable a range of new investigations in neuroscience: First, it will make it possible to efficiently optimize detailed single-cell models. This will allow insights into cellular properties across cell types and their relationship with, for example, transcriptomic measurements[39–41], as well as into the contribution of dendritic processing to neural computations[28].

Second, it will facilitate creating large-scale biophysical network models. Such network models[4] have so far primarily been built in a bottom-up fashion. JAXLEY opens up possibilities to train thousands of parameters with gradient descent, allowing the tuning of large biophysical networks as a whole.

Third, in addition to training such models, JAXLEY will also enable numerous other applications, such as gradient-based Bayesian inference, the analysis of stability via Lyapunov exponents, adversarial attacks, the computation of maximally excitable stimuli or the design of optimally discriminative experiments.

A central challenge in training biophysical models to perform computational tasks will be to bridge the timescales between biophysical mechanisms (milliseconds) and behavior (seconds). We showed that it is possible to fit data of up to 1 s in length, but optimizing models on tasks that require backpropagating gradients along even longer simulations is challenging. In addition, the loss landscape of biophysical models can be highly non-convex, which could trap optimization in local minima[42]. Gradient descent also requires the definition of a single loss function to be optimized, which can cause challenges when different tasks or features of experimental data are captured best by different parameter sets[43]. To investigate how gradient-based optimization could accommodate such scenarios, we have designed JAXLEY such that it can be combined with any scheme for updating parameters. For example, for some tasks and training paradigms, forward mode automatic differentiation[44] or evolutionary algorithms[16,21] have been reported to perform similarly to (or sometimes even better than) backprop. JAXLEY directly supports GPU-accelerated implementations of either of these algorithms, opening up possibilities to develop new

methods for training of biophysical neural systems. By building upon the framework JAX[20], Jaxley will further benefit from advances in the deep learning community at scaling and training large simulations. Finally, as the parameter space grows, there will typically be a wide range of solutions that match experimental recordings, and a single 'best' fit might not be representative of the full solution space in the presence of parameter degeneracy[2]. To address such situations, Jaxley could be combined with methods that aim to identify the full space of data-compatible models, such as Bayesian inference[17], or its parallelization capabilities could be exploited for parameter scans[2].

New experimental tools allow scientists to measure connectivity, morphology, genetic identity and activity of neural circuits at increasing levels of detail and scale. Jaxley is a powerful tool that can integrate measurements of connectivity and morphology into biophysical simulations, while allowing the resulting networks to be fitted to data or computational tasks—just like deep neural networks. This will enable investigations of the biophysical basis of neural computation at unprecedented scales.

## Online content

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

## Methods

### Solving differential equations of biophysically detailed single-neuron models

We used an implicit Euler solver for the voltage equations, and an exponential Euler solver for the gates (Supplementary Note 2). We simulated voltage at internal nodes and, similar to Neuron, also simulated terminal nodes at branch points. To solve the tridiagonal system of equations of every branch, we used the algorithm presented by Stone[45].

### Robust training of biophysical models

We used the following tools to improve training accuracy and robustness. First, we used parameter transformations that ensure that optimization of bounded parameters can be performed in unconstrained space and that biophysical parameters are on the same scale. In particular, we used an inverse sigmoid transformation $T(\theta) = -\log\left(\frac{1}{(\theta-l)/(u-l)} - 1\right)$, where $l$ is the lower bound and $u$ is the upper bound. Second, we used a variant of Polyak stochastic gradient descent[23], which computes the step as $\text{step} = \gamma \frac{\nabla\mathcal{L}}{||\nabla\mathcal{L}||^\beta}$. This optimizer overcomes large variations of the gradient between steps. We used values of $\beta \in [0.8, 0.99]$. In some applications, we further multiplied the gradient with the loss value $\mathcal{L}$ (ref. 23), such that the optimizer automatically reduced the learning rate toward the end of training. Third, when necessary, we used different optimizers for different types of parameters. This affects the Polyak gradient descent optimizer because it normalizes by the gradient norm. Fourth, we performed multilevel checkpointing to reduce the memory requirements of backpropagation through time. Fifth, when we observed vanishing or exploding gradients, we performed truncated backpropagation through time. We used this only for the hybrid mechanistic/statistical model of the retina. When performed, we interrupted gradient computation every 50 ms. All of these tools are implemented in Jaxley. We provide an overview of training schemes for all examples in Supplementary Note 3.

Jaxley can be combined with any parameter optimization or learning method. The ability to evaluate the gradient with backprop opens possibilities for training biophysical systems with gradient descent or with gradient-based reinforcement learning methods, but Jaxley can also be efficiently combined with gradient-free parameter optimization methods (for example, genetic algorithms, Bayesian optimization, gradient-free reinforcement learning approaches or biologically plausible learning rules such as Hebbian learning). In these cases, Jaxley can parallelize forward simulations whose output can inform the gradient-free update rule.

### Fitting single-cell models to voltage recordings

**Active mechanisms.** We used two types of sodium, five types of potassium, two types of calcium and a hyperpolarization-activated channel and inserted them in soma, basal and apical dendrites, and axon, following Van Geit et al.[16].

**Parameter bounds and initialization.** We optimized 19 parameters. These were the same parameters as used by Van Geit et al.[16], but our model did not contain a persistent sodium channel in the axon. We used the same parameter search bounds as Van Geit et al.[16] for fitting the L5PC to synthetic data (Fig. 2b), but we enforced that somatic potassium existed (lower bound 0.25 S cm$^{-2}$). For the experimental recordings from the Allen Cell Types Database, we used slightly larger bounds for three of the parameters to increase the flexibility of the model: We used bounds of [0, 6] S cm$^{-2}$ (instead of [0, 4] S cm$^{-2}$) for somatic and axonal sodium channels and a lower bound of 1 ms for the delay of the calcium buffer (instead of 20 ms). In addition, we made the following six parameters trainable: To account for the fact that channel models were built upon measurements from other cells, we allowed a fivefold variation of the time constant of the high-voltage-activated and low-voltage-activated calcium channels and the M channel, and

we introduced a parameter that can shift the activation curve of the somatic sodium channel by up to 10 mV to the left. In addition, to account for ion channels typically being selective to more than a single ion, we allowed the potassium reversal potential to vary within [−100, −70] mV and the sodium reversal potential to vary within [40, 60] mV. To fit recordings from the Allen Cell Types Database (IDs 485574832, 488683425, 480353286 and 473601979), we modified the leak conductance to 10$^{-4}$ for all cells, the leak reversal potential to −88, −88, −95 and −95 mV and capacitance to 6, 2, 6 and 2.5 μF cm$^{-2}$, respectively.

**Summary statistics and loss functions.** Optimizing parameters of biophysical models with gradient descent requires that the loss function, and therefore also the summary statistics, are differentiable. This is not the case for typically used features such as spike count, which led us to defining different loss functions.

For the synthetic data (Fig. 2b), we split the voltage trace into two windows and used the mean and standard deviation of these two windows as summary statistics. This led to a total of four summary statistics. To standardize the data, we divided the mean voltages by 8.0 and standard deviations by 4.0. We used a mean absolute error loss on the standardized summary statistics.

To fit data from the Allen Cell Types Database, we used AllenSDK to download morphologies and electrophysiology. We used a loss function based on DTW. Unlike the standard Euclidean (pointwise) distance, which can overly penalize minor temporal misalignment of fast events such as action potentials, DTW actively aligns the two signals in time before measuring their pointwise differences. This alignment emphasizes the geometric similarity between time series rather than strict temporal correspondence, which we found to work well for fitting spiking voltage traces. Because the standard DTW discrepancy is not fully differentiable, we use the soft DTW discrepancy[46]. For efficient computation on GPUs, we use a custom implementation of soft DTW discrepancy in JAX. We preprocessed the raw voltage traces by applying a sliding window maximum reduction with a window size of 50 time steps and a stride of 30 time steps (each time step is 0.025 ms). We rescaled both the observations and simulations to the unit interval using the same scaling factors. We used the cost function $c(x_i, y_i) = |x_i - y_i| + |i - j|$, where $i$ and $j$ are time points and $x$ and $y$ are voltage traces.

**Training procedure.** For the synthetic data, we used our variant of Polyak gradient descent with $\gamma = 1/3$ and $\beta = 0.8$. We trained for ten optimization steps. We also ran the genetic algorithm (IBEA-DEAP as implemented in the BluePyOpt package, version 1.14.11)[16] for ten iterations, with ten simulations per iteration.

For models based on the Allen Cell Types Database, we fitted the first 200 ms of the trace with a loss based on dynamic time warping (see above) and Polyak gradient descent with $\gamma = 5 \times 10^{-4}$ and $\beta = 0.99$. For the fits with large computational budget (Fig. 2d), we parallelized 1,000 gradient descent runs on a GPU and trained for 50 steps. After training, we simulated the final 1,000 parameter sets for 1,150 ms and selected the best parameter set as the one minimizing the error across the following five features: overall spike count, and spike amplitude and spike timing of the first two spikes (spike amplitude was multiplied with a weight of 1/3 for computing the total error). We followed the same procedure for the genetic algorithm (first fit 200 ms with dynamic time warping and then selected the best set from the final population). We ran the genetic algorithm with 1,000 offspring (which we also parallelized on GPU) for 50 generations. For the fits with reduced computational budget (Extended Data Fig. 5), we followed the same procedure and used the same hyperparameters, but we used only ten gradient descent runs and trained only for ten steps.

### Fitting voltage recordings of all branches

**Active mechanisms.** We used the same ion channels as for the task of fitting electrophysiology traces. To generate the observation, we

sampled from a Gaussian process the conductance profiles as a function of the Euclidean distance from the soma. We did this for the maximal conductances of all active ion channels in the apical dendrite (three active mechanisms) and the axon (seven active mechanisms).

**Parameter bounds and initialization.** For the active mechanisms in the apical dendrite and the axon, we defined one parameter per branch in the cell and initialized parameters randomly and independently from each other. We used a single parameter for each mechanism in the soma and basal dendrite. This resulted in a total of 1,390 parameters. Parameters had the same lower and upper bounds as for the L5PC described above.

**Training procedure.** We used a mean absolute error loss function between the observed voltages and the simulated voltages, evaluated at every fifth time step between 1 ms and 5 ms of simulation time. We used our custom optimizer with $\beta = 0.8$, momentum of 0.9 and learning rate of 0.1. We trained the system for 500 iterations. We regularized the optimization such that neighboring branches had similar conductance values. In particular, we added to the loss the term $\lambda \cdot \sum_{b=1}^{\text{branches}} (\bar{g}_b - \text{parent}(\bar{g}_b))^2$, with regularization strength $\lambda = 0.001$.

**Bayesian inference.** To perform Bayesian inference over the parameters, we used a uniform prior over the free parameters with equivalent bounds as with the original training procedure. We also incorporated the same regularizer into the prior distribution.

We sampled from the posterior distribution with Hamiltonian Monte Carlo, which leverages the available gradient to efficiently sample from high-dimensional distributions. We initialized 200 chains with samples from the prior distribution. Each chain was run for 500 iterations in parallel on a single NVIDIA GeForce A100 GPU. For each iteration, we performed five integration steps for the Hamiltonian dynamics with a step size of 0.01, leading to an average acceptance rate of ≈65%. The last sample of each chain was used as a posterior sample. We used the BlackJAX[47] implementation of Hamiltonian Monte Carlo.

To visualize the learned conductance profile, we discretized distance from soma into 11 bins and grouped all parameters within each bin. We then calculated histograms for each of the bins and generated a spline interpolation of all quantile lines (Fig. 2h).

### Nonlinear single-neuron computation
**Active mechanisms.** The model contained sodium, potassium and leak channels in all branches, with dynamics following the default implementation of the NEURON package[12].

**Parameter bounds and initialization.** We trained sodium, potassium, and leak maximal conductances, as well as radius, length, and axial resistivity of every compartment in the model. Parameters were initialized randomly within uniform bounds. The bounds were [0.05, 1.1] for sodium, [0.01, 0.3] for potassium and [0.0001, 0.001] for leak, all in S cm$^{-2}$. The bounds for the radius were [0.1, 5.0] μm, [1, 20] μm per compartment for the length and [500, 5,500] Ωcm for the axial resistivity.

**Training procedure.** We used a mean absolute error loss function between the simulated voltages after 3 ms of simulated time and the class label (35 mV or −70 mV). We used the Adam optimizer with a learning rate of 0.01, and a batch size of 1. We trained the system for 200 epochs. One iteration of gradient descent took 1.7 ms on an Apple MacBook Pro CPU.

### Hybrid model of the retina
**Data, model and active mechanisms.** Ran et al.[18] presented images of 15 × 20 pixels to a mouse retina. Each pixel had a size of 30 × 30 μm. Each image was presented for 200 ms, and a total of 1,500 images were presented. The images were centered on recording fields of the calcium

activity. We used the first 1,024 images of these as the training dataset. In total, 15 recording fields were made for the Off alpha cell used in our work, leading to a total of 1,024 × 15 training data points. Within each recording field, Ran et al.[18] defined a variable number of regions of interest, within which the calcium activity was recorded. In total, the data contained 232 regions of interest.

To denoise the calcium data, we low-pass filtered the raw calcium data with a Butterworth filter and a cutoff frequency of 7 Hz. We z-scored the resulting signal, with a different mean and standard deviation for each region of interest. Next, we generated a single label for each image. We did this by performing a linear regression from the image onto delayed calcium signals, and then used the calcium value at the delay, which was most predictive (that is, had the highest Pearson correlation between the linear regression prediction and data). This led us to a delay of 1.8 s. As the label, we used the low-pass filtered calcium value after this delay (starting from image onset). We followed Ran et al.[18] to compute receptive fields. We used automatic smoothness detection[48] with 20 iterations of evidence optimization. We standardized all receptive fields to range from 0 to 1 and, for contours, thresholded the receptive fields at a value of 0.6.

We modeled photoreceptors as a spatial linear Gaussian filter. The filter had a standard deviation of 50 μm. We modeled bipolar cells as point neurons with a nonlinearity. The point neurons were spaced on a hexagonal grid with a distance between neurons of 40 μm. The nonlinearity was taken from values measured by Schwartz et al.[29]. We connected every bipolar cell onto every branch of the RGC that was within 20 μm of the bipolar cell. Within every branch of the RGC, the synapse was made to the compartment that had the minimal Euclidean distance to the bipolar cell.

We computed the calcium activity of the model as the intracellular calcium value of the compartment that was the closest to the experimental recording site. We convolved this value with a double-exponential kernel to model dynamics of the calcium indicator. We used a rise time of 5 ms and decay time of 100 ms.

Our model of the RGC contained six ion channels, which were developed based on measurements from the RGCs in cats[49]. The channels were a sodium channel, a leak channel, a delayed-rectifier potassium channel, a transient potassium channel, a calcium-dependent potassium channel and a calcium channel. The model had all of these channels in every compartment of the model.

**Parameter bounds and initialization.** We initialized all membrane conductances at previously published values[49], except for sodium conductances, which we initialized at 0.15 S cm$^{-2}$ in the soma and 0.05 S cm$^{-2}$ in the dendrite. We sampled the initial synaptic strengths randomly within 0 and 0.1 nS, and then divided the synaptic conductance by the number of postsynaptic connections that a bipolar cell makes (such that, in expectation, every bipolar cell has the same impact on the RGC). We initialized the axial resistivity of every compartment at 5,000 Ωcm. Finally, we initialized the radius of every dendritic compartment at 0.2 μm. We kept the diameter of the soma constant at 10 μm.

We trained the following set of parameters: One value for each maximal conductance in the soma (six parameters), one value for each maximal conductance in the dendrites, shared across all dendrites (six parameters), one value for each branch radius (147 parameters), one value for the axial resistivity of each branch (147 parameters) and one value for each synaptic conductance from the bipolar cells onto the RGCs (250 parameters). In total, the model had 556 parameters.

We used the following bounds for optimization: For somatic conductances, we used [0.05, 0.5] for sodium, [0.01, 0.1] for potassium, [$10^{-5}, 10^{-3}$] for leak, [0.01, 0.1] for transient potassium, [$2 \times 10^{-5}, 2 \times 10^{-4}$] for calcium-dependent potassium and [0.002, 0.003] for calcium. All membrane conductance units are S cm$^{-2}$. For dendritic conductances, we used the same bounds apart from the lower bound of 0 for sodium. For the branch radii, we used bounds of [0.1, 1.0] μm (ref. 18). For the

axial resistivities, we used [100, 10,000] Ωcm. For the synaptic conductances, we used [0.0, 0.2] nS.

**Training procedure.** We trained the model with Polyak stochastic gradient descent with momentum. We considered every kind of parameter (somatic conductance, dendritic conductance, radii, axial resistivities and synaptic conductances) as separate parameters (which influences the computation of the gradient norm in Polyak stochastic gradient descent). We trained the model for ten epochs with a learning rate of 0.01 and a momentum of 0.5. We used $\beta = 0.99$ to compute the norm in Polyak stochastic gradient descent. We used a batch size of 256 and used two-level checkpointing to reduce the memory of backpropagation. To avoid vanishing or exploding gradients, we truncated the gradient in time. Specifically, we reset the gradient to 0 after 50, 100 and 150 ms of simulations.

We trained the model with mean absolute error loss between the experimentally measured (low-pass filtered) calcium value and the model-predicted calcium value after 200 ms. Recording sites for which no recording was available in the data were masked out in the loss computation.

**Inductive bias.** To evaluate the inductive bias of the hybrid model, we trained several such models, each on a reduced dataset. We reduced the dataset by using only a subset of stimulus–recording pairs from one scan field. We repeated this procedure over seven scan fields and five sizes of stimulus–recording pairs, namely 32, 64, 128, 256 and 512. We trained each model with stochastic gradient descent as described above. To ensure that the gradient is stochastic, we used batch sizes of 4, 4, 8, 16 and 32 for the five dataset sizes, respectively. We performed early stopping based on a validation set that contained 25% of the training dataset and evaluated performance on 512 test data points. To avoid exceedingly long training times, we trained for at most 100 steps.

For the artificial neural network, we used a multilayer perceptron with three hidden layers of [100, 100, 50] units and with ReLU activation functions.

### An RNN of biophysical neurons performing an evidence integration task
**Active mechanisms and synapses.** All recurrent units used the same ion channels as for the task involving the L5PC, analogously inserted in soma, basal and apical dendrites. The two readout neurons were passive. The synapses were conductance based, as described by Abbott and Marder[50].

The recurrent units were connected with a probability of 0.2 by synapses from the soma of the presynaptic neurons to the apical dendrites of the postsynaptic neurons. All recurrent units had synapses onto the two readout neurons. Around 50% of recurrent units had inhibitory outgoing synapses, created by setting their synaptic reversal potential to −75 mV, and the other half were excitatory with a synaptic reversal potential of 0 mV.

The RNNs in Fig. 4b had 50 units with an 80% excitatory/20% inhibitory split, and a connection probability of 0.2.

**Parameter bounds and initialization.** We trained the maximal synaptic conductances and the weights of the stimulus input to each neuron in the network. We used bounds of [−0.2, 0.2] for the input weights and [0, 3 max($g$)] for the maximal synaptic conductances. In total, the model had 109 trainable parameters.

Initial values for the maximum synaptic conductances were drawn from a standard normal distribution scaled by an initial gain $g$ such that the bulk of the eigenspectrum of the synaptic weights lay in a circle on the complex plane with radius $g$ (after multiplying inhibitory synapse weights by −1). We presented inputs by stimulating neurons at their basal dendrites. We set $g = 5.0\pi/5 \times 10^3$, which is close to the transition point between stable and chaotic dynamics. For recurrent connections, we set the rate constant for transmitter–receptor dissociation rate

(the $k_-$ parameter[2]), that influences the synaptic time constants, to 1/1.0 ms. For connections to the rate-based readout neurons, we used slower synapses, with $k_-$ set to 1/40 ms. All recurrent units received stimulus input scaled by random initial weights drawn from $\mathcal{N}(0, 0.1)$.

**Stimulus generation.** Stimuli were generated by sampling values from normal distributions and then low-pass filtering them with a maximum frequency cutoff of 2,500 Hz. For training, we sampled values from $\mathcal{N}(\mu_-, 0.05)$ and $\mathcal{N}(\mu_+, 0.05)$, where $\mu_- \sim \mathcal{N}(-0.005, 0.0002)$ and $\mu_+ \sim \mathcal{N}(0.005, 0.0002)$.

**Training procedure.** We used a batch size of four, three levels of checkpointing, the Adam optimizer with a learning rate of 0.01, 2,000 gradient steps and gradient normalization with $\beta = 0.8$. We sampled new training data at every gradient step. Sweeps were used to inform the choice of hyperparameters. We used a cross-entropy loss function with logits calculated as the mean readout activities in the last 20 ms of stimulus presentation. The training time was 7 h on an Intel(R) Xeon(R) Silver 4116 CPU @ 2.10 GHz.

### An RNN of biophysical neurons performing a delayed-match-to-sample task
**Active mechanisms and synapses.** We used the same channel mechanisms, compartments and synapses as in the evidence integration task. We used a network of 50 units, with a recurrent connectivity probability of 0.05, and all units connecting to the readout neurons. Approximately 80% and 20% of the neurons had excitatory and inhibitory outgoing connections, respectively.

**Parameter bounds and initialization.** We trained the maximal synaptic conductances, constrained to the range $[0, \infty)$, using a SoftPlus function. We also trained the weights of the stimulus input to each neuron in the network, restricted to the range [0, 4], as well as $k_-$ restricted to [0.05, 2]. This resulted in 459 parameters.

We set $g = 5.0\pi/5 \times 10^3$. We set the $k_-$ parameter of recurrently connected neurons to 1, and to 0.1 for the slower synapses onto the readout neurons. The connection probability from input to (the basal dendrite of) recurrent units was 0.1, with initial weights drawn from $\mathcal{U}_{[0,1]}$.

**Stimulus generation.** Stimuli consisted of square pulses with additive Gaussian noise sampled from $\mathcal{N}(0, 0.001)$. The onset period was 20 ms, and the stimulus and response durations were 50 ms. The delay period changed throughout the training procedure. Initial delay durations were drawn uniformly from $\mathcal{U}_{[50,150]}$. The average delay duration was increased in steps of 100 ms, whenever the network got at least 95% accuracy on a single batch, until $\mathcal{U}_{[450,550]}$ was reached.

We used a batch size of 64, two levels of checkpointing and the Adam optimizer with a learning rate of 0.001, which was exponentially decayed to 0.0001 over 1,000 epochs. We used a cross-entropy loss function for each time step, computed during the response period of the task. The training time was around 10 h on an NVIDIA GeForce RTX 3090.

**Computing Lyapunov exponents.** We quantified the stability of recurrent networks by the maximal Lyapunov exponent. To obtain the maximal Lyapunov exponent, we discretized our model to obtain $\mathbf{x}_{t+1} = \mathbf{f}(\mathbf{x}_t)$, where $\mathbf{x}_t$ contains voltages and gating variables at time $t$, and $\mathbf{f}$ is one step of the solver. We used the following numerical algorithm to approximate $\lambda_{\max}(\mathbf{x}_0)$[33]: First, we generated an initial state $\mathbf{x}_0$ and initial unit norm vector $\mathbf{q}_0$. After discarding initial transients for 4 s of simulation, we let the system $\mathbf{x}_{t+1} = \mathbf{f}(\mathbf{x}_t)$ and $\mathbf{q}_{t+1} = D\mathbf{f}|_{\mathbf{x}_t}\mathbf{q}_t$ (where $D$ denotes the Jacobian) evolve for $T = 2.4 \times 10^5$ time steps (a further 6 s). Note that $\mathbf{q}_{t+1}$ can be efficiently computed using Jacobian vector products in JAX[20]. At every time step we computed $r_t = \|\mathbf{q}_t\|$ and renormalized: $\mathbf{q}_t \leftarrow \frac{\mathbf{q}_t}{r_t}$. The maximal Lyapunov exponent was then given by $\frac{1}{T}\sum_1^T \log \| r_t \|$.

**A biophysical network that performs computer vision tasks**
**Active mechanisms and synapses.** The first layer consisted of 28 × 28 neurons, each stimulated by a step current whose amplitude was proportional to a pixel value. Each neuron in the first layer had a ball-and-stick morphology: Each cell consisted of four compartments, where one compartment (the soma) had a radius of 10 μm and all other compartments had a radius of 1 μm and a length of 10 μm. The input and hidden layer contained sodium, potassium and leak channels in all branches, with dynamics following the default implementation of the NEURON package[12]. The output layer consisted of ten neurons with ball-and-stick morphologies (such as the input layer neurons) and with leak dynamics. We used conductance-based synapses as described by Abbott and Marder[50]. The layers of the network were densely connected.

**Parameter bounds and initialization.** We optimized sodium, potassium and leak maximal conductances of every branch in the network (55,000 parameters) and all synaptic conductances (51,000 parameters). We used the following bounds for the parameters: $[0.05, 0.5]$ for sodium, $[0.01, 0.1]$ for potassium, $[0.0001, 0.001]$ for leak, $[-5 / 28^2 / 25, 5 / 28^2 / 25]$ for synapses from the input to the hidden layer and $[-5 / 64 / 25 / 2, 5 / 64 / 25 / 2]$ for synapses from the hidden layer to the output layer. We initialized sodium maximal conductances at $0.12$ S cm$^{-2}$, potassium at $0.036$ S cm$^{-2}$ and leak at $0.0003$ S cm$^{-2}$. We initialized synaptic conductances as samples from a Gaussian distribution with a mean of 0 and standard deviation of $1 / 28^2 / 25$ for the first layer and a standard deviation of $1 / 64 / 25 / 2$ for the second layer. We set the synaptic rate constant for transmitter–receptor dissociation to $k_- = 1/4$.

**Training procedure.** We used a batch size of 16 and cross-entropy loss based on the values $(v + 65)/3$, where $v$ is the somatic voltage of the output neurons after 10 ms of simulation. We used a cosine learning rate schedule and trained the network for seven epochs. Each gradient step took 25 s on a V100 GPU.

**Adversarial attacks.** To perform the adversarial attacks, we performed optimization of the input with gradient descent. We normalized every gradient step and used a learning rate of 5.0. We used bounds of $[0, 1]$ for all pixel values during optimization. We used a cross-entropy loss function. We chose the target label for the adversarial attack randomly, but ensured that the target label is not the true label. We computed accuracy based on 128 adversarial attacks.

**Reporting summary**
Further information on research design is available in the Nature Portfolio Reporting Summary linked to this article.

## Data availability
All code to generate results and figures is available at https://github.com/mackelab/jaxley_experiments/. We used AllenSDK (v2.16.2) to obtain data from the Allen Cell Types Database (IDs 485574832, 488683425, 480353286 and 473601979). We downloaded CA1 neurons (Figs. 1d–f and 5) from https://neuromorpho.org/ (the cell shown in Fig. 1d has ID NMO_00120, other cells are from the same archive and of the same cell type). We downloaded the L5PC (Fig. 2) from BluePyOpt (v1.14.11) and converted the ASC file to SWC with the morph-tool software. We used JAX (v0.4.29), NumPy (v1.26.4), pandas (v2.2.1) and BlackJAX (v1.1.0) for data analysis and simulation.

## Code availability
Our toolbox, JAXLEY, is openly available at https://github.com/jaxleyverse/jaxley/ under the Apache 2.0 license. Tutorials and examples of how to use the toolbox are available at https://jaxley.readthedocs.io/en/latest/. A collection of channels and synapses for use with JAXLEY is available at https://github.com/jaxleyverse/jaxley-mech/ under the MIT license.

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

## Acknowledgements
We thank N. Bosch, P. Hennig, S. Müller, S. Axen, T. Euler, M. Bethge, T. Zenkel, F. D'Agostino, J.-M. Lueckmann and all members of our research groups for discussions. This work was supported by the German Research Foundation (DFG) through Germany's Excellence Strategy (EXC 2064 - project number 390727645; to P.B. and J.H.M.) and the Collaborative Research Center 1233 'Robust Vision' (to P.B. and J.H.M.), SFB 1089 (PN 227953431; to J.H.M.), the 'Certification and Foundations of Safe Machine Learning Systems in Healthcare' project funded by the Carl Zeiss Foundation (to J.H.M.) and the European Union (ERC, 'DeepCoMechTome', ref. 101089288; to J.H.M., 'NextMechMod', ref. 101039115; to P.B.). Views and opinions expressed are those of the authors only and do not necessarily reflect those of the European Union or the European Research Council Executive Agency. Neither the European Union nor the granting authority can be held responsible for them. M.D., K.L.K., J.B., M.P., M.G. and J.K.L. are members of the International Max Planck Research School for Intelligent Systems (IMPRS-IS).

## Author contributions
Conceptualization and methodology: M.D., P.B., P.J.G. and J.H.M. Software and investigation: M.D., K.L.K., M.P., J.B., M.G., Z.H., J.K.L. and C.S. Analysis: M.D., K.L.K., M.P., P.B., P.J.G. and J.H.M. Writing: M.D., K.L.K., M.P., P.B., P.J.G. and J.H.M. Writing (review and editing): J.B., Z.H., M.G., J.K.L. and C.S. Funding acquisition: P.B. and J.H.M. Supervision: P.B., P.J.G. and J.H.M.

## Funding

## Competing interests
The authors declare no competing interests.

## Additional information
**Extended data** is available for this paper at https://doi.org/10.1038/s41592-025-02895-w.

**Correspondence and requests for materials** should be addressed to Michael Deistler or Jakob H. Macke.

Peer reviewer reports are available. Primary Handling Editor: Nina Vogt, in collaboration with the *Nature Methods* team.

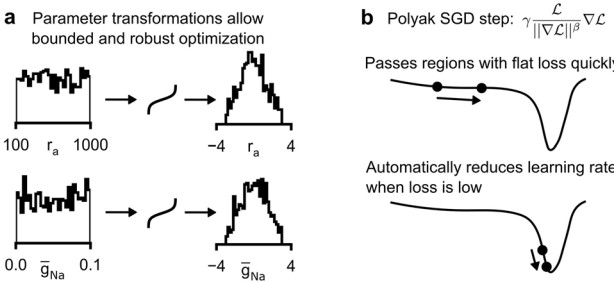

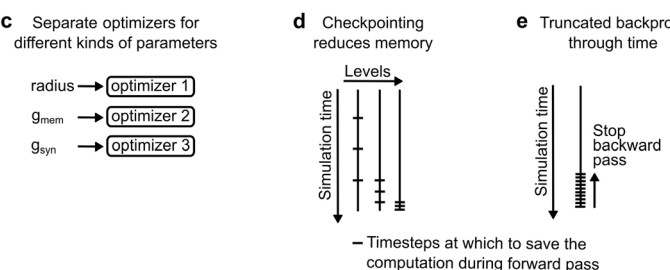

**Extended Data Fig. 1 | Robust and efficient gradient descent.** (**a**) Histogram of two biophysical parameters (rows, random samples within optimization bounds) before (left) and after (right) parameter transformations. The transformation is designed such that the parameters are unconstrained and of the same scale. (**b**) Illustrative loss surface and update step made by our variant of Polyak gradient descent. (**c**) We use different optimizers for different kinds of parameters. (**d**) Illustration of multi-level checkpointing. We use checkpointing to overcome memory limitations of backpropagation of error. Multi-level checkpointing requires multiple forward passes per gradient computation, but typically reduces memory requirements. (**e**) Illustration of truncated backpropagation through time. Truncated backpropagation through time allows to overcome vanishing or exploding gradients at the cost of providing only an approximate gradient.

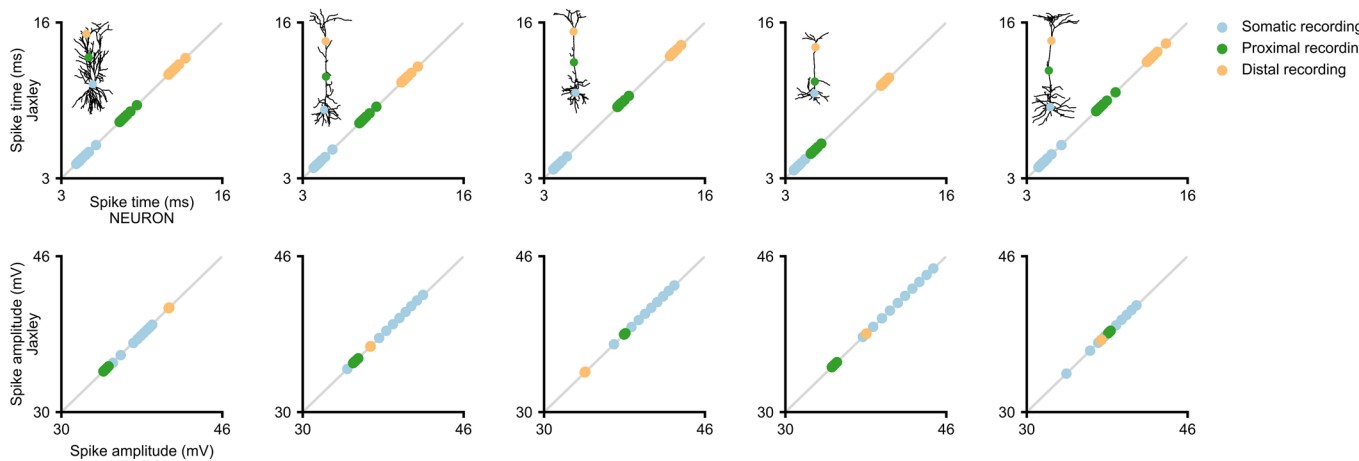

**Extended Data Fig. 2 | Accuracy of the implicit solver in Jaxley.** Top: Spike time at three recording sites (somatic, proximal, distal) for input stimuli of 10 different amplitudes, ranging from 0.2 nA to 1.1 nA (individual dots). Columns are different morphologies from the Allen Cell Types Database. Bottom: Same as top, for spike amplitude.

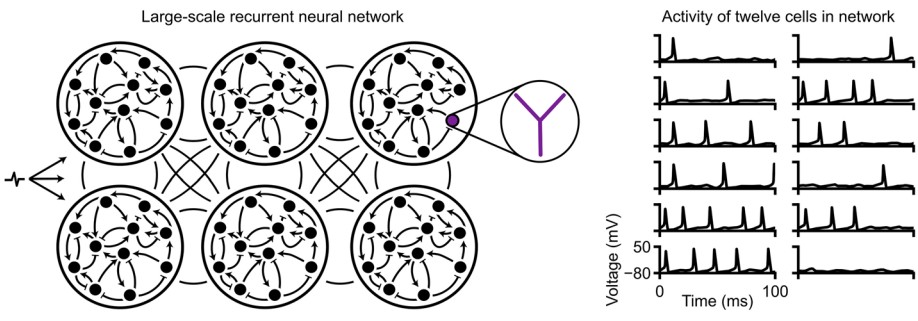

**Extended Data Fig. 3 | Network of simplified neurons with millions of synapses.** The networks consist of 12k neurons, each with three branches and Hodgkin–Huxley dynamics. Networks were connected densely within clusters and sparsely between clusters via tanh-synapses, for a total of 25.4 million synapses. Jaxley can simulate networks of this complexity on a single A100 GPU (activity on the right) and also compute the gradient with respect to more than 25 million parameters (all sodium, potassium, leak, and synaptic conductances) with backpropagation. For this example, we used the average activity of twelve randomly selected neurons at the last timepoint to compute the loss.

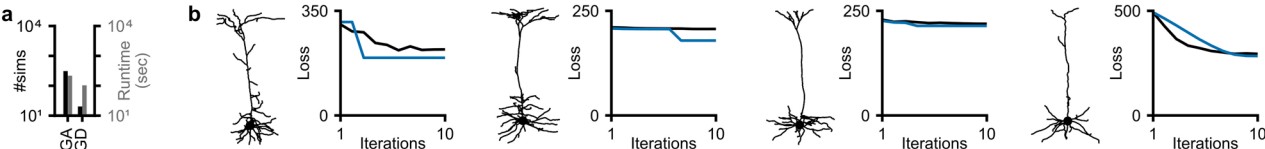

**Extended Data Fig. 4 | Fitting single-cell models.** (**a**) L5PC model. Number of simulations and runtime required to achieve a loss smaller than 0.55 when fitting the L5PC to synthetic recordings, with genetic algorithm (GA) and gradient descent (GD). Averaged across ten runs with 10 iterations each. (**b**) Allen Cell Types models. Loss over the number of iterations for the median of ten genetic algorithms, each run with ten simulations per iterations (black) and the minimal loss across ten gradient descent runs (blue). Ten gradient descent runs require the same number of simulations as a single run of the genetic algorithm. Across all four morphologies, gradient descent identifies parameter sets with lower loss than the genetic algorithm, but we note that computing the gradient incurs additional computational cost.

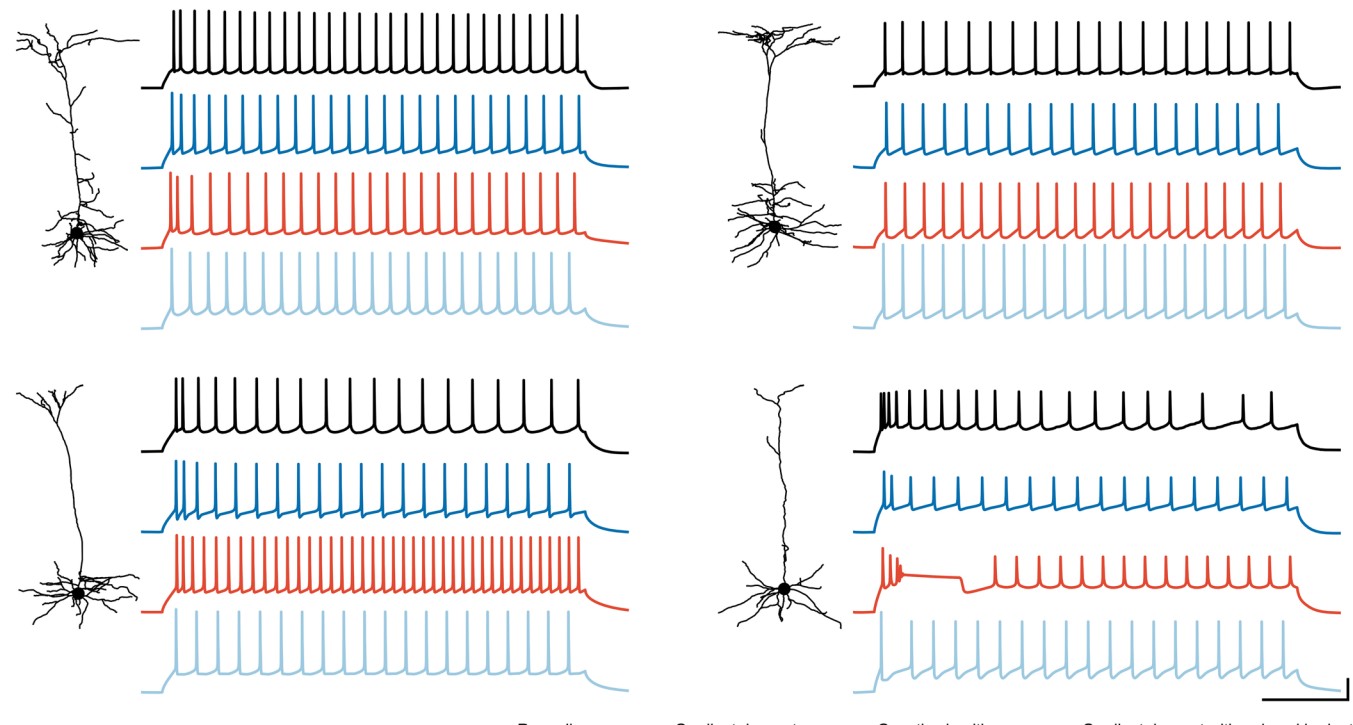

———— Recording    ———— Gradient descent    ———— Genetic algorithm    ———— Gradient descent with reduced budget

**Extended Data Fig. 5 | Recordings from the Allen Cell Types Database (black) as well as gradient descent (blue) and genetic algorithm (red) fits.** Gradient descent and the genetic algorithm were run for fifty iterations and with 1,000 simulations per step (in parallel on GPU). The gradient descent fit with reduced budget (light blue) was run for ten iterations and ten simulations per step. Scalebars: 200 ms and 30 mV.

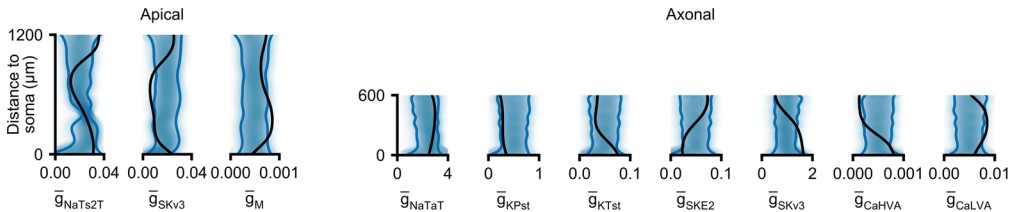

**Extended Data Fig. 6 | Bayesian inference of membrane conductances.** We used Jaxley with gradient-based Hamiltonian Monte–Carlo to infer the posterior distribution over membrane conductances of a layer 5 pyramidal cell. Blue lines are 90 % confidence intervals, black line is the ground truth that was used to generate the synthetic observation.

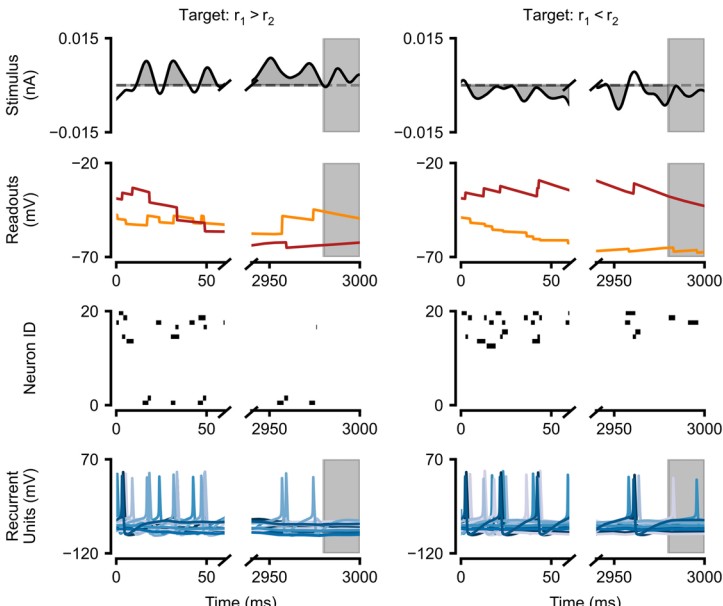

**Extended Data Fig. 7 | Generalization of the evidence integration task.** Evidence integration task performance on three seconds of stimulus with positive integral (left) and negative integral (right). We used the same network parameters as in Fig. 4.

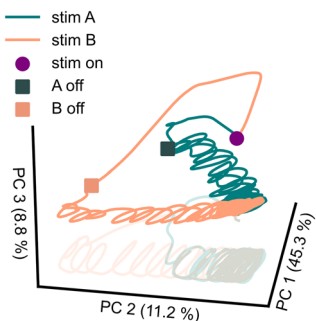

**Extended Data Fig. 8 | Long-term dynamics reveal transient coding.**
Long-term dynamics of the network from Fig. 4 trained to perform the
delayed-match-to-sample task. The figure shows the dynamics of the full state of
the model (including ion concentrations), projected into principal component
space, after briefly presenting either stimulus. We first smoothed each dynamic
variable with a Hann window of 40 ms, and normalised, before computing the
principal components.

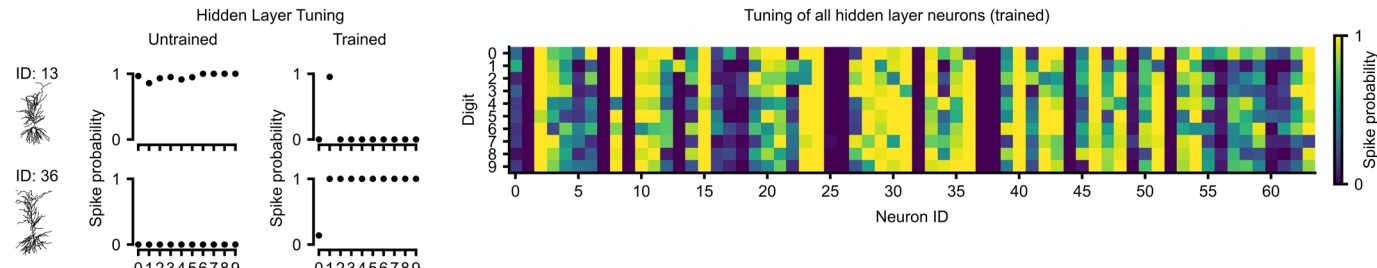

**Extended Data Fig. 9 | Hidden layer tuning of the biophysical network classifying MNIST.** Left: Before training, the two shown cells were untuned, but they developed tuning to specific digits (ON-tuning for digit '1' for ID 13 and OFF-tuning for digit '0' for ID 36) after training. Right: Hidden layer tuning of all neurons after training. We evaluated the fraction of images in the dataset for which each hidden neuron spiked.

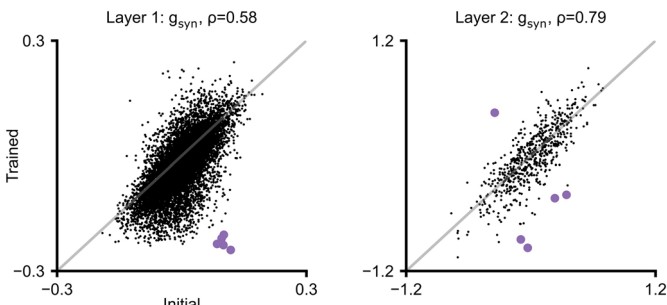

**Extended Data Fig. 10 | Initial vs trained synaptic conductances for synapses in the first (left) and second layer (right).** Purple dots highlight the five most changed synaptic conductances. Especially for the first layer, synaptic conductances changed strongly during training. In both layers, parameters were significantly correlated to their initial value ($p < 10^{-8}$, two-sided t-test).

|  | |
|---|---|

# Reporting Summary

## Statistics

For all statistical analyses, confirm that the following items are present in the figure legend, table legend, main text, or Methods section.

| n/a | Confirmed | |
|---|---|---|
| ☐ | ☒ | The exact sample size ($n$) for each experimental group/condition, given as a discrete number and unit of measurement |
| ☒ | ☐ | A statement on whether measurements were taken from distinct samples or whether the same sample was measured repeatedly |
| ☐ | ☒ | The statistical test(s) used AND whether they are one- or two-sided<br>*Only common tests should be described solely by name; describe more complex techniques in the Methods section.* |
| ☒ | ☐ | A description of all covariates tested |
| ☐ | ☒ | A description of any assumptions or corrections, such as tests of normality and adjustment for multiple comparisons |
| ☐ | ☒ | A full description of the statistical parameters including central tendency (e.g. means) or other basic estimates (e.g. regression coefficient) AND variation (e.g. standard deviation) or associated estimates of uncertainty (e.g. confidence intervals) |
| ☐ | ☒ | For null hypothesis testing, the test statistic (e.g. $F$, $t$, $r$) with confidence intervals, effect sizes, degrees of freedom and $P$ value noted<br>*Give P values as exact values whenever suitable.* |
| ☐ | ☒ | For Bayesian analysis, information on the choice of priors and Markov chain Monte Carlo settings |
| ☒ | ☐ | For hierarchical and complex designs, identification of the appropriate level for tests and full reporting of outcomes |
| ☒ | ☐ | Estimates of effect sizes (e.g. Cohen's $d$, Pearson's $r$), indicating how they were calculated |

*Our web collection on statistics for biologists contains articles on many of the points above.*

## Software and code

Policy information about availability of computer code

| Data collection | We used AllenSDK (v2.16.2) to obtain data from the Allen Cell Types Database (IDs 485574832, 488683425, 480353286, 473601979).<br><br>We downloaded CA1 neurons (Fig. 1d,f, Fig. 5) from Neuromorpho.org (the cell shown in Fig. 1d has ID NMO_00120, other cells are from the same archive and of the same cell type).<br><br>We downloaded the L5PC (Fig. 2) from BluePyOpt (v1.14.11, https://github.com/BlueBrain/BluePyOpt/blob/master/examples/l5pc/morphology/C060114A7.asc). We converted this file to SWC with the morph-tool software (https://github.com/BlueBrain/morph-tool). |
|---|---|
| Data analysis | We developed JAXLEY to perform biophysical simulations: https://github.com/jaxleyverse/jaxley<br>For the results shown in the paper, we used JAXLEY version 0.1.2.<br><br>Channel models are implementeed in our Jaxley-Mech library: https://github.com/jaxleyverse/jaxley-mech<br>For the results shown in the paper, we used version 0.1.0.<br><br>Experiments shown in the paper can be reproduced with the code provided here: https://github.com/mackelab/jaxley_experiments<br><br>We used JAX (v0.4.29), NumPy (v1.26.4), pandas (v2.2.1), and BlackJAX (v1.1.0) for data analysis and simulation. |

For manuscripts utilizing custom algorithms or software that are central to the research but not yet described in published literature, software must be made available to editors and reviewers. We strongly encourage code deposition in a community repository (e.g. GitHub). See the Nature Portfolio guidelines for submitting code & software for further information.

## Data

Policy information about availability of data

All manuscripts must include a data availability statement. This statement should provide the following information, where applicable:
- Accession codes, unique identifiers, or web links for publicly available datasets
- A description of any restrictions on data availability
- For clinical datasets or third party data, please ensure that the statement adheres to our policy

> We used only publicly available data from the Allen Cell Types Database (https://celltypes.brain-map.org) and from Ran et al. (2020, available on Zenodo: https://zenodo.org/records/3708064). We used morphologies that are publicly available on NeuroMorpho (https://neuromorpho.org).

## Research involving human participants, their data, or biological material

Policy information about studies with human participants or human data. See also policy information about sex, gender (identity/presentation), and sexual orientation and race, ethnicity and racism.

| | |
|---|---|
| Reporting on sex and gender | Not applicable. |
| Reporting on race, ethnicity, or other socially relevant groupings | Not applicable. |
| Population characteristics | Not applicable. |
| Recruitment | Not applicable. |
| Ethics oversight | Not applicable. |

Note that full information on the approval of the study protocol must also be provided in the manuscript.

# Field-specific reporting

Please select the one below that is the best fit for your research. If you are not sure, read the appropriate sections before making your selection.

☒ Life sciences ☐ Behavioural & social sciences ☐ Ecological, evolutionary & environmental sciences

For a reference copy of the document with all sections, see nature.com/documents/nr-reporting-summary-flat.pdf

# Life sciences study design

All studies must disclose on these points even when the disclosure is negative.

| | |
|---|---|
| Sample size | For panel 3f, a total of seven independent datasets were used to compare the two architectures. No statistical methods were used to pre-determine this sample size. No other results performed statistical tests. |
| Data exclusions | No data was excluded. |
| Replication | For panel 3f, model training and evaluation were independently replicated across the seven datasets. Each dataset served as an independent test of model performance. A single statistical comparison (one-sided t-test) was performed across these seven runs. |
| Randomization | For panel 3f, datasets were selected independently of model performance and not influenced by experimental outcome. |
| Blinding | Blinding was not applicable to panel 3f, as model training and evaluation were fully automated. |

# Reporting for specific materials, systems and methods

We require information from authors about some types of materials, experimental systems and methods used in many studies. Here, indicate whether each material, system or method listed is relevant to your study. If you are not sure if a list item applies to your research, read the appropriate section before selecting a response.

## Materials & experimental systems

| n/a | Involved in the study |
|-----|----------------------|
| ☒ ☐ | Antibodies |
| ☒ ☐ | Eukaryotic cell lines |
| ☒ ☐ | Palaeontology and archaeology |
| ☒ ☐ | Animals and other organisms |
| ☒ ☐ | Clinical data |
| ☒ ☐ | Dual use research of concern |
| ☒ ☐ | Plants |

## Methods

| n/a | Involved in the study |
|-----|----------------------|
| ☒ ☐ | ChIP-seq |
| ☒ ☐ | Flow cytometry |
| ☒ ☐ | MRI-based neuroimaging |

## Plants

| | |
|---|---|
| Seed stocks | Not applicable. |
| Novel plant genotypes | Not applicable. |
| Authentication | Not applicable. |

