## [Peer Review file · Nature Methods]

JAXLEY: Differentiable simulation enables large-scale training of detailed biophysical models of neural dynamics

Corresponding Author: Mr Michael Deistler

Version 0:

Decision Letter:

19th Dec 2024

Dear Dr Deistler,

Thank you for your patience. Your Article, "Differentiable simulation enables large-scale training of detailed biophysical models of neural dynamics", has now been seen by two reviewers. As you will see from their comments below, although the reviewers find your work of considerable potential interest, they have raised a number of concerns. We are interested in the possibility of publishing your paper in Nature Methods, but would like to consider your response to these concerns before we reach a final decision on publication. We therefore invite you to revise your manuscript to address these concerns.

Link Redacted

We hope to receive your revised paper within 2-3 months. If you cannot send it within this time, please let us know. In this event, we will still be happy to reconsider your paper at a later date so long as nothing similar has been accepted for publication at Nature Methods or published elsewhere.

OPEN SCIENCE REQUIREMENTS

REPORTING SUMMARY AND EDITORIAL POLICY CHECKLISTS

EXTENDED DATA FIGURES

DATA AVAILABILITY

All novel DNA and RNA sequencing data, protein sequences, genetic polymorphisms, linked genotype and phenotype data, gene expression data, macromolecular structures, and proteomics data must be deposited in a publicly accessible database, and accession codes and associated hyperlinks must be provided in the "Data Availability" section.

CODE AVAILABILITY

Please include a "Code Availability" subsection in the Online Methods which details how your custom code is made available. Only in rare cases (where code is not central to the main conclusions of the paper) is the statement "available upon request" allowed (and reasons should be specified).

MATERIALS AVAILABILITY

ORCID

Nature Methods is committed to improving transparency in authorship. As part of our efforts in this direction, we are now requesting that all authors identified as 'corresponding author' on published papers create and link their Open Researcher and Contributor Identifier (ORCID) with their account on the Manuscript Tracking System (MTS), prior to acceptance. This applies to primary research papers only. ORCID helps the scientific community achieve unambiguous attribution of all scholarly contributions. You can create and link your ORCID from the home page of the MTS by clicking on 'Modify my Springer Nature account'. For more information please visit <http://www.springernature.com/orcid>.

Best regards,
Nina

Nina Vogt, PhD
Senior Editor
Nature Methods

Reviewers' Comments:

Reviewer #1 (Remarks to the Author):

As a computational neuroscientist for over 20 years, I am confident that the methods developed here are a major breakthrough both in terms of simulating biophysically detailed models of neurons and neuronal circuits, and as a way to optimize the models both to accurately replicate experimental data, and further to train detailed neuronal models to perform different types of behavioral tasks. As such, I would recommend having this manuscript published, with only minor modifications, and additional suggested tests below.

Although impressive, including more difficult tasks would highlight the strengths of the authors' approach. I would recommend the authors test their optimization algorithms for neuronal network circuit models, on other tasks similar to the following where models need to perform more complex behaviors in environments with dynamically changing variables, such as video game play, or navigation, etc. as illustrated in other recent papers:
<https://doi.org/10.3389/fncom.2022.1017284>
<https://doi.org/10.1371/journal.pone.0265808>

Another point of discussion is whether the methods illustrate anything about how real neuronal circuits learn or have been optimized through evolution. This would be important to highlight given that the authors found many of the parameter values/ranges did not shift substantially on average, between trained and untrained models, so perhaps the difference between functional and dysfunctional circuits is only marginal. Therefore, additional clarification on the parameter distributions, whether particular neurons are more prominent in solving a specific task, would be important to understand how the modeled circuits learn to perform behaviors appropriately.

The software in the github repository is clear, easy to read, and provides a large amount of documentation,

and tutorials making it straightforward for the community to use the valuable tools the authors developed.

It will be useful to benchmark simulations on different hardware architectures, since this will motivate people to try the new software, depending on their hardware capabilities.

I would like some clarification/discussion on how the simulation software compares against other approaches/libraries for simulating neuronal circuits such as BINDSnet (<https://doi.org/10.3389/fninf.2018.00089>) and NetPyNE (<https://doi.org/10.7554/eLife.44494>) ?

Some details on: Why are Izhikevich and integrate and fire neurons not included in the software library? These simplified (compared to HH) neurons are widely used. I would suggest including basic implementations of at least adaptive integrate-and-fire neurons in future versions of Jaxley.

Are synaptic weights in a fairly large circuit model modifiable with the GD optimization? What about optimization of architectural changes (number of neurons, number of layers) ? Can Jaxley optimize synaptic "noise" inputs, which are often required to maintain activity in a circuit model?

Does Jaxley support standard learning rules like hebbian plasticity, STDP, STDP/RL ? Those are important and widely used and it would be interesting to compare whether those rules can help with any of the proposed tasks.

Can Jaxley optimize parameters of intracellular dynamics, like calcium signaling (molecular interactions and reaction-diffusion) ? Is that supported or will it be supported in the future? Those types of dynamics are generally considered important to neuronal cellular function, but are often excluded from multiscale neural models that perform behaviors. Partly, that's due to difficulty in optimizing and simulating such models. Jaxley could provide a solution. I would suggest the authors could comment on these questions.

How difficult would it be to optimize the type of model presented here (<https://doi.org/10.1016/j.celrep.2023.113378>) in Jaxley to replicate firing rates and physiological dynamics? These types of models are now routinely constructed, but were written in other simulators (NEURON). So, can the NEURON modelers hope to benefit from Jaxley in the near term?

Reviewer #1 (Remarks on code availability):

Clear code with documentation, tutorials, example models provided in a github repository

Reviewer #2 (Remarks to the Author):

Summary of the key results

This manuscript presents JAXLEY, a novel simulator for detailed biophysical neuron models and networks thereof, built upon the open-source JAX library, developed by Google for high-performance numerical computing and machine learning.

By leveraging the capabilities of JAX for automatic differentiation, just-in-time compilation, and ease of parallelization using GPUs and TPUs (Tensor Processing Units), JAXLEY opens up the possibility of performing very rapid gradient-based optimisation across very large numbers of parameters, with applications to fitting models to data, and training networks of biologically-detailed neurons to perform useful tasks.

JAXLEY provides a Python-based programming environment that will be familiar to users of other biophysical neuron simulation tools, notably NEURON.

Originality and significance

Many other authors have built neural simulators on top of machine learning frameworks (PyTorch or Tensorflow), for example:

- Norse (<https://norse.github.io/norse/>; Pehle & Pederson, 2021)
- BindsNET (<https://bindsnet-docs.readthedocs.io/>; Hazan et al., 2018)
- snnTorch (<https://snntorch.readthedocs.io/>; Eshraghian et al., 2023)
- NengoDL/KerasSpiking (<https://www.nengo.ai/nengo-dl/>; <https://www.nengo.ai/keras-spiking/>)

As far as I am aware, all of these tools support mainly, or only, point-neuron models (e.g., integrate-and-fire (I&F) family, Izhikevich), and are focused on implementing traditional deep learning approaches using spiking neural networks.

Other researchers have targeted biophysical neural models, but not through directly solving the models' differential

equations, rather by using recurrent neural networks to generate/predict the future activity of the neuron/network (Oláh et al., 2022; Wang et al., 2022).

Given this context of previous work, the principal points of originality of the current work appear to me as follows:

1. application of automatic gradient computation to biophysical, multi-compartment models, rather than to point neuron, I&F-type models;
2. a focus on fitting models to experimental data, alongside the more traditional deep learning capability of training networks to perform tasks;
3. possibly, implementation of the implicit Euler method in a machine learning framework (I have not investigated which integration methods are implemented by the tools cited above);
4. implementation on top of JAX, which apparently has generally better performance than PyTorch or Tensorflow.

While each of these points individually is a valuable advance, taken together they make the current study highly significant. While previous ML-based simulation tools have focused mainly on a machine learning audience, JAXLEY has the potential to transform the practice of computational neuroscience, opening up avenues for investigation that could not be explored previously due to lack of computational resources. This is not to discount its potential value in machine learning approaches, studying for example the potential benefits of neuronal non-linearities in the computational capabilities and efficiency of deep and/or recurrent networks.

Data & methodology: validity of approach, quality of presentation, clarity and context

I have successfully installed the JAXLEY package, but due to time limitations have not been able to thoroughly test it, or to inspect the code base in detail. With these provisos, the implementation appears to be very solid, actively developed, and following good practices such as the use of continuous integration testing.

While the results presented in this manuscript make me wish to try JAXLEY in a real world project, I am underwhelmed by the quality of the fitting results shown for single neurons. The fits in the left-panel of Figure 2d, and all of the panels in Supplementary Figure S3 are poor: the response of the fitted model does not capture the initial short burst, and the adaptation seen in the right-hand model in S3 is in the opposite direction to that seen in the experimental data (decreasing rather than increasing inter-spike intervals). This suggests either (i) a poor choice of loss function or (ii) the fitting has got stuck in a local minimum. I do not think the fits shown are a good advertisement for the capabilities of the software.

The writing and figures are generally clear and well presented, although I found the individual panels within the figures to be very small: I had to zoom in a lot to properly view them. There are some formatting problems in the references, with many cases of uppercase letters converted to lowercase (for example NEST->Nest, NEURON->neuron, BluePyOpt->Bluepyopt, GPU->gpu), suggesting use of BibTeX without appropriate escaping (by surrounding letters that should stay uppercase with curly braces {}).

Conclusions

In conclusion, this is a very exciting advance for biologically detailed modelling and simulation in neuroscience, potentially game-changing if it lives up to its promise. The work is technically solid, distributed under an open-source licence, and open to contributions. The work is well presented, with a number of mostly-convincing use cases/demonstrations (but see comments above on single neuron fitting results).

Suggested improvements

I would suggest the following improvements:

1. A discussion of previous work on spiking neuron simulators built on top of machine learning frameworks (Norse, BindsNET, snnTorch, etc.), and of how the approach used in JAXLEY relates to/goes beyond those tools.
2. More discussion of the limitations/challenges of gradient descent in neural modelling, for example:
 - (i) The findings by Prinz and Marder that disparate mechanisms can give rise to very similar network activity (e.g., Prinz et al., 2004, which the manuscript cites but does not discuss), which challenge the idea of a "best fit".
 - (ii) The finding that the parameter spaces of complex neuron models are highly non-smooth, with narrow "valleys" or wormholes in parameter space connecting widely separated regions of good fit (e.g., Achard & De Schutter, 2006).
 - (iii) The concept of multi-objective optimisation, i.e. the observation that when fitting to multiple loss functions there may be a family of good models following a "Pareto front" (e.g., Druckmann et al., 2007; Jedlicka et al., 2022).
 - (iv) The common problem of getting stuck in local minima, and the associated need to add noise, for example through simulated annealing. It would be interesting to know if JAXLEY already provides functionality for this, or if this is a potential extension.

I would also like more information on the "specific optimizers for non-convex loss surfaces", which it seems to me would be important for many problems in computational neuroscience.

3. If this can be done in a reasonable amount of time, produce better fits for Figs 2d and S3 that reproduce the qualitative features seen in the experimental traces.
4. Fix the upper/lower-case problems in the reference list.

References

- Pehle Christian and Pedersen Jens Egholm (2021). Norse - A deep learning library for spiking neural networks, Zenodo, <https://doi.org/10.5281/zenodo.4422025>
- Hazan Hananel, Saunders Daniel J., Khan Hassaan, Patel Devdhar, Sanghavi Darpan T., Siegelmann Hava T. and Kozma Robert (2018), BindsNET: A Machine Learning-Oriented Spiking Neural Networks Library in Python, *Frontiers in Neuroinformatics* 12:89 <https://doi.org/10.3389/fninf.2018.00089>
- Eshraghian Jason K, Ward Max, Neftci Emre, Wang Xinxin, Lenz Gregor, Dwivedi Girish, Bennamoun Mohammed, Jeong Doo Seok and Lu Wei D. (2023) Training spiking neural networks using lessons from deep learning. *Proceedings of the IEEE* 111:1016--1054
- Oláh VJ, Pedersen NP and Rowan MJM (2022) Ultrafast simulation of large-scale neocortical microcircuitry with biophysically realistic neurons *eLife* 11:e79535. <https://doi.org/10.7554/eLife.79535>
- Wang T, Wang Y, Shen J, Wang L and Cao L (2022) Predicting spike features of hodgkin-huxley-type neurons with simple artificial neural network *Frontiers in Computational Neuroscience* 15:800875. <https://doi.org/10.3389/fncom.2021.800875>
- Achard P, De Schutter E (2006) Complex Parameter Landscape for a Complex Neuron Model. *PLoS Comput Biol* 2(7): e94. <https://doi.org/10.1371/journal.pcbi.0020094>
- Druckmann S, Banitt Y, Gidon A, Schürmann F, Markram H, Segev I. A novel multiple objective optimization framework for constraining conductance-based neuron models by experimental data. *Front Neurosci.* 2007 Oct 15;1(1):7-18. doi: 10.3389/neuro.01.1.1.001.2007.
- Jedlicka Peter, Bird Alexander D. and Cuntz Hermann (2022). Pareto optimality, economy–effectiveness trade-offs and ion channel degeneracy: improving population modelling for single neurons. *Open Biol.* 12:220073 <http://doi.org/10.1098/rsob.220073>
- Prinz, A., Bucher, D. & Marder, E. Similar network activity from disparate circuit parameters. *Nat Neurosci* 7, 1345–1352 (2004). <https://doi.org/10.1038/nn1352>

Review by Andrew P. Davison (<http://orcid.org/0000-0002-4793-7541>)

Reviewer #2 (Remarks on code availability):

I have successfully installed the JAXLEY package, but due to time limitations have not been able to thoroughly test it, or to inspect the code base in detail. With these provisos, the implementation appears to be very solid, actively developed, and following good practices such as the use of continuous integration testing.

Author Rebuttal letter:

Dear Dr. Vogt,

We thank you and the reviewers for the constructive and detailed comments and for appreciating the importance of our work. We are excited about the assessment of our work as a “major breakthrough” (R1), “highly significant” (R2), that it “has potential to transform the practice of computational neuroscience, opening up avenues for investigation that could not be explored previously” (R2), and that one reviewer explicitly recommends publication (R1). We are particularly glad that the reviewers recognize our efforts in developing and maintaining JAXLEY as a user-friendly software, calling the software “clear, easy to read, and [...] straightforward for the community to use” (R1), as well as “very solid, actively developed, and following good practices” (R2).

Based on the comments of the reviewers, we performed several new analyses and included multiple improved explanations. In particular, we now

- provide an overview with differences to other simulators (R1 & R2),
 - show improved single-neuron fits to experimental data (R2, see new Supp. Fig. S5),
 - benchmark JAXLEY on additional compute platforms (R1, see new Supp. Fig. S3),
 - demonstrate that JAXLEY can simulate a network of 12k neurons with simplified morphologies and 25M synapses on a single GPU (R1, see new Supp. Fig. S4).
- In addition, based on the recommendations of the reviewers, we have extended the JAXLEY toolbox by adding the two additional features (including tutorials on how to use these features):

- support for ion diffusion (R1),
- support for simulating simplified neuron models (e.g., LIF, Izhikevich; R1).

We hope that our new results and modifications further strengthen the manuscript and are grateful for the reviewers’ constructive suggestions.

Sincerely, and on behalf of all authors

1/12

Reviewer #1 (Remarks to the Author):

As a computational neuroscientist for over 20 years, I am confident that the methods developed here are a major breakthrough both in terms of simulating biophysically detailed models of neurons and neuronal circuits, and as a way to optimize the models both to accurately replicate experimental data, and further to train detailed neuronal models to perform different types of behavioral tasks. As such, I would recommend having this manuscript published, with only minor modifications, and additional suggested tests below.

Thank you for the positive feedback! We hope that JAXLEY opens new possibilities to build biophysical models and to tune them to fit experimental data or solve computational tasks.

Although impressive, including more difficult tasks would highlight the strengths of the authors' approach. I would recommend the authors test their optimization algorithms for neuronal network circuit models, on other tasks similar to the following where models need to perform more complex behaviors in environments with dynamically changing variables, such as video game play, or navigation, etc. as illustrated in other recent papers:

<https://doi.org/10.3389/fncom.2022.1017284>, <https://doi.org/10.1371/journal.pone.0265808>

Indeed, using gradient descent to optimize models on complex behavioral tasks is a major contribution of JAXLEY. Optimizing biophysical models on tasks as the one shown in these papers should be possible in principle. However, investigating highly challenging reinforcement learning tasks, like the ones proposed in these papers, and in particular in biophysically detailed models, would be beyond the scope of this paper, and likely requires full, separate studies. In general, however, JAXLEY can efficiently be combined with arbitrary learning paradigms, which we now highlight in the updated manuscript.

Another point of discussion is whether the methods illustrate anything about how real neuronal circuits learn or have been optimized through evolution. This would be important to highlight given that the authors found many of the parameter values/ranges did not shift substantially on average, between trained and untrained models, so perhaps the difference between functional and dysfunctional circuits is only marginal.

We do believe that Jaxley will enable neuroscientists to build more realistic mechanistic models which will shed light on learning in biological circuits. We note, however, that the gradient descent algorithms that JAXLEY uses to optimize models are not biologically plausible. Several recent studies have aimed to find and evaluate biologically plausible variants of, or alternatives to, backpropagation [1,2,3]. All of these approaches focused on networks with very simple neuron models. JAXLEY opens up opportunities for investigating these questions also in mechanistically realistic circuits. We have added the sentence "The ability to perform gradient descent may also provide insights into how biological systems learn or are optimized." to discuss this.

A second approach to studying this question, and as suggested by the reviewer, would be to inspect how the learning process shifts parameter values in mechanistic networks. While our simulations were not specifically set up to target this question, we nevertheless note that they show that—even with the same learning rule—different tasks lead to different results in this respect:

For the MNIST task, the distribution of parameters does indeed not change much (Fig. 5e), but individual parameters can change largely (but trained parameters remain correlated to

2/12

their initial values, matching findings in artificial neural networks [4]). This further strengthens our claim that the aggregate statistics of connectivity parameters in networks may not be sufficient to generate functional activity. We include this analysis in Supp. Fig. S11 of the updated manuscript:

In contrast, we also show that, on other tasks, the distribution of parameters can change strongly: for example, on the evidence-integration task with RNNs (Fig. 4f).

Therefore, additional clarification on the parameter distributions, whether particular neurons are more prominent in solving a specific task, would be important to understand how the modeled circuits learn to perform behaviors appropriately.

We agree that understanding how trained networks solve tasks will be an important area of research. We have begun to investigate such questions in Fig. 5g in the manuscript, where we inspected the tuning of hidden neurons in a task-trained network and found that some neurons code for individual digits after training.

The software in the github repository is clear, easy to read, and provides a large amount of documentation, and tutorials making it straightforward for the community to use the valuable tools the authors developed.

Thank you for appreciating our efforts in making JAXLEY a usable toolkit. We are committed to maintaining it and to making it the go-to toolbox for biophysical neuron modelling.

It will be useful to benchmark simulations on different hardware architectures, since this will motivate people to try the new software, depending on their hardware capabilities. We have added benchmarks on two new hardware architectures: A MacBook Pro M3 CPU and a NVIDIA H100 GPU. The MacBook Pro was almost twice as fast as the previously used AMD CPU. The NVIDIA H100 had lower simulation time than an NVIDIA A100 and was twice as fast for very large batches of parameter sets processed in parallel. We have added the following figure to the updated manuscript, showing these results:

3/12

I would like some clarification/discussion on how the simulation software compares against other approaches/libraries for simulating neuronal circuits such as BINDSnet

(<https://doi.org/10.3389/fninf.2018.00089>)

and NetPyNE (<https://doi.org/10.7554/eLife.44494>) ?

Thank you for suggesting this! We have added a discussion to the updated manuscript (new Methods section "Comparison to other toolboxes").

Some details on: Why are Izhikevich and integrate and fire neurons not included in the software library? These simplified (compared to HH) neurons are widely used. I would suggest including basic implementations of at least adaptive integrate-and-fire neurons in future versions of Jaxley.

We thank the reviewer for raising this. Indeed, in the original manuscript, Izhikevich and Integrate-and-fire neurons were not yet implemented as part of JAXLEY (due to its focus on detailed biophysical models). We have now added an implementation for (Leaky) integrate-and-fire neurons and for Izhikevich neurons (and we have clarified the statement on the documentation website). At the moment, JAXLEY does not yet implement and test surrogate gradient methods that can efficiently optimize these models (because of their spike discontinuity, direct gradients are not applicable), but we are planning to add this feature in the future.

Are synaptic weights in a fairly large circuit model modifiable with the GD optimization? What about optimization of architectural changes (number of neurons, number of layers) ? Can optimize synaptic "noise" inputs, which are often required to maintain activity in a circuit model?

JAXLEY can optimize synaptic weights in (very) large circuit models. As we show in Fig. 1f, JAXLEY can differentiate a circuit with 2k morphologically detailed neurons and 1M synapses, and in Fig. 5 we show that JAXLEY can optimize 50k synapses to fit MNIST. Following your question below, in a new Supp. Fig. S4, we now also show that JAXLEY can simulate and differentiate a circuit of 12k neurons with simplified morphologies (with Hodgkin-Huxley dynamics) and connected by 25M synapses, on a single GPU.

JAXLEY can also optimize synaptic "noise" inputs. In Fig. 5f, we show that JAXLEY can optimize input currents adversarially to "fool" the network into wrong classifications. Other optimization paradigms regarding (synaptic) inputs are also possible (e.g., to find maximally excitatory images [1]).

Architectural changes, however, cannot easily be optimized with gradient descent, as these are typically discrete and thus non-differentiable parameters (e.g., number of hidden neurons).

4/12

As such, optimizing architectures requires different optimizers such as evolutionary algorithms. JAXLEY supports arbitrary learning or update rules and can be used to implement such algorithms for optimizing architecture.

Does Jaxley support standard learning rules like hebbian plasticity, STDP, STDP/RL ? Those are important and widely used and it would be interesting to compare whether those rules can help with any of the proposed tasks.

Yes, JAXLEY flexibly supports diverse update rules. In building JAXLEY, we have ensured that it is a modular framework: It can simulate networks and compute their gradients with respect to inputs or parameters. How (or if) these gradients are used, however, is fully up to the user. Gradients can be plugged into a reinforcement learning algorithm, or they could be fully ignored in, e.g., a Hebbian learning paradigm. We believe that this flexibility of JAXLEY is one of its major strengths and will enable the community to develop and evaluate new powerful training methods for neural circuits. We clarify these abilities of JAXLEY in the manuscript.

Can Jaxley optimize parameters of intracellular dynamics, like calcium signaling (molecular interactions and reaction-diffusion) ? Is that supported or will it be supported in the future? Those types of dynamics are generally considered important to neuronal cellular function, but are often excluded from multiscale neural models that perform behaviors. Partly, that's due to difficulty in optimizing and simulating such models. Jaxley could provide a solution. I would suggest the authors could comment on these questions.

Indeed, JAXLEY can model and optimize parameters of intracellular dynamics. In Fig. 2a-e, we optimized the decay time constant of the calcium buffer and the fraction of free calcium, and any other factor of intracellular dynamics could, in principle, also be optimized.

Yet, at the time of submission of this manuscript, JAXLEY had not yet supported reaction-diffusion dynamics. To make even more detailed simulations possible, we have now implemented ion diffusion. The diffusion strength and ion dynamics can be optimized with gradient descent. We have written a tutorial on using these features on our documentation website (https://jaxley.readthedocs.io/en/latest/tutorials/11_ion_dynamics.html) and also mention this possibility in the manuscript.

How difficult would it be to optimize the type of model presented here (<https://doi.org/10.1016/j.celrep.2023.113378>) in Jaxley to replicate firing rates and physiological dynamics?

JAXLEY can optimize circuits of that scale on a single GPU. As a proof-of-principle demonstration, we have now set up a network of 12k neurons with three compartments each and connected by 25 million synapses. On one NVIDIA A100 GPU, running 100 ms of simulation took around 15 seconds, and computing the gradient with respect to all cellular conductances and all 25 million synaptic weights took 130 seconds. We have included this analysis as Supplementary Fig. S4 in the updated manuscript:

5/12

These types of models are now routinely constructed, but were written in other simulators (NEURON). So, can the NEURON modelers hope to benefit from Jaxley in the near term? We agree that lowering barriers for usage is important for adoption of JAXLEY. We have already built a prototype for an automated tool which converts ion channel models from .nmodl (used by NEURON) to Python (such that they can be read by JAXLEY) and will release this tool soon. We are in discussions with the Allen Institute to provide support for using JAXLEY with the SONATA file format they use. We are in contact with several labs which are switching from NEURON to JAXLEY in order to find out how to support such transitions most effectively, and we are organising a tutorial on the upcoming CNS conference to reach more potential users. Overall, we hope that the computational power of JAXLEY, combined with our efforts to easily port and construct models, will allow NEURON users to easily transition to JAXLEY.

Reviewer #1 (Remarks on code availability):

Clear code with documentation, tutorials, example models provided in a github repository

References:

[1] Ahmad, N., van Gerven, M. A., & Ambrogioni, L. (2020). Gait-prop: A biologically plausible learning rule derived

from backpropagation of error. *Advances in Neural Information Processing Systems*, 33, 10913-10923.

[2] Bengio, Y., Lee, D. H., Bornschein, J., Mesnard, T., & Lin, Z. (2015). Towards biologically plausible deep learning. arXiv preprint arXiv:1502.04156.

[3] Bartunov, S., Santoro, A., Richards, B., Marris, L., Hinton, G. E., & Lillicrap, T. (2018). Assessing the scalability of biologically-motivated deep learning algorithms and architectures. *Advances in neural information processing systems*, 31.

[4] Chizat, L., Oyallon, E., & Bach, F. (2019). On lazy training in differentiable programming. *Advances in neural information processing systems*, 32

[5] Walker, Edgar Y., et al. "Inception loops discover what excites neurons most using deep predictive models." *Nature neuroscience* 22.12 (2019): 2060-2065.

6/12

Reviewer #2 (Remarks to the Author):

Summary of the key results

This manuscript presents JAXLEY, a novel simulator for detailed biophysical neuron models and networks thereof, built upon the open-source JAX library, developed by Google for high-performance numerical computing and machine learning.

By leveraging the capabilities of JAX for automatic differentiation, just-in-time compilation, and ease of parallelization using GPUs and TPUs (Tensor Processing Units), JAXLEY opens up the possibility of performing very rapid gradient-based optimisation across very large numbers of parameters, with applications to fitting models to data, and training networks of biologically-detailed neurons to perform useful tasks.

JAXLEY provides a Python-based programming environment that will be familiar to users of other biophysical neuron simulation tools, notably NEURON.

Originality and significance

Many other authors have built neural simulators on top of machine learning frameworks (PyTorch or Tensorflow), for example:

- Norse (<https://norse.github.io/norse/>; Pehle & Pederson, 2021)

- BindsNET (<https://bindsnet-docs.readthedocs.io/>; Hazan et al., 2018)

- snnTorch (<https://snntorch.readthedocs.io/>; Eshraghian et al., 2023)

- NengoDL/KerasSpiking (<https://www.nengo.ai/nengo-dl/>; <https://www.nengo.ai/keras-spiking/>)

As far as I am aware, all of these tools support mainly, or only, point-neuron models (e.g., integrate-and-fire (I&F) family, Izhikevich), and are focused on implementing traditional deep learning approaches using spiking neural networks.

Other researchers have targeted biophysical neural models, but not through directly solving the models' differential equations, rather by using recurrent neural networks to generate/predict the future activity of the neuron/network (Oláh et al., 2022; Wang et al., 2022). Given this context of previous work, the principal points of originality of the current work appear to me as follows:

1. application of automatic gradient computation to biophysical, multi-compartment models, rather than to point neuron, I&F-type models;
2. a focus on fitting models to experimental data, alongside the more traditional deep learning capability of training networks to perform tasks;
3. possibly, implementation of the implicit Euler method in a machine learning framework (I have not investigated which integration methods are implemented by the tools cited above);
4. implementation on top of JAX, which apparently has generally better performance than PyTorch or Tensorflow.

While each of these points individually is a valuable advance, taken together they make the current study highly significant. While previous ML-based simulation tools have focused mainly on a machine learning audience, JAXLEY has the potential to transform the practice of computational neuroscience, opening up avenues for investigation that could not be explored previously due to lack of computational resources. This is not to discount its potential value in

7/12

machine learning approaches, studying for example the potential benefits of neuronal non-linearities in the computational capabilities and efficiency of deep and/or recurrent networks. Thank you for appreciating the significance of our work! We would like to highlight the third point in your list of contributions: Implementing the implicit Euler method for morphologically

detailed single neuron models requires efficiently solving a sparse linear system defined by the branching structure of a neuron (Hines, 1984). Implementing such a solver in a machine learning framework (and doing this in an efficiently parallelizable way, following, e.g., Ben-Shalom et al., 2022) is indeed a major contribution of our work.

Data & methodology: validity of approach, quality of presentation, clarity and context

I have successfully installed the JAXLEY package, but due to time limitations have not been able to thoroughly test it, or to inspect the code base in detail. With these provisos, the implementation appears to be very solid, actively developed, and following good practices such as the use of continuous integration testing. Thank you for this positive feedback! We are committed to maintaining JAXLEY and to making it the go-to tool for biophysical neuron modelling.

While the results presented in this manuscript make me wish to try JAXLEY in a real world project, I am underwhelmed by the quality of the fitting results shown for single neurons. The fits in the left-panel of Figure 2d, and all of the panels in Supplementary Figure S3 are poor: the response of the fitted model does not capture the initial short burst, and the adaptation seen in the right-hand model in S3 is in the opposite direction to that seen in the experimental data (decreasing rather than increasing inter-spike intervals). This suggests either (i) a poor choice of loss function or (ii) the fitting has got stuck in a local minimum. I do not think the fits shown are a good advertisement for the capabilities of the software.

We agree that the fits shown in Fig. 2d and S3 are not optimal. In these fits, we had prioritized a proof-of-concept demonstration with low computational budget (roughly 2 minutes per cell on CPU).

To demonstrate that gradient descent can achieve closer fits, we have now made six additional parameters trainable, we modified the loss function, and we have scaled up fitting on a GPU. By running on GPU, we could run 1k gradient descent chains in parallel, allowing for a much better exploration of the parameter space. This procedure took 1-2h per neuron, and the resulting fits are shown below. We now achieve an excellent fit for three out of four cells, and a significantly improved fit for the last cell. Exploration of more complex fitting methods (e.g., by letting the gradient descent chains interact) or loss functions could further improve the fits. Using JAXLEY, we also parallelized a genetic algorithm on GPU and found that – for these comparably simple models – it identified fits of similar quality. Finally, we note that these fits are significantly better than the best biophysical models provided on the Allen Cell-Types Database. However, the results are difficult to compare as the Allen-Cell-Types Database uses a different set of ion channels and optimizes its cells across a range of protocols. As such, we do not include the comparison to the Allen fits in the updated manuscript (but the rest of the figure is now available as Supplementary Fig. S6).

8/12

The writing and figures are generally clear and well presented, although I found the individual panels within the figures to be very small: I had to zoom in a lot to properly view them. We used font size 6pt for all figures, which is within the recommendations of the Nature submission guide (5-7pt). Nonetheless, we agree that some panels in Fig. 2 had been relatively small, and we have adapted this figure (panels d,e,f,g).

There are some formatting problems in the references, with many cases of uppercase letters converted to lowercase (for example NEST->Nest, NEURON->neuron, BluePyOpt->Bluepyopt, GPU->gpu), suggesting use of BibTeX without appropriate escaping (by surrounding letters that should stay uppercase with curly braces {}). Thank you for pointing this out. We have fixed all references.

Conclusions

9/12

In conclusion, this is a very exciting advance for biologically detailed modelling and simulation in neuroscience, potentially game-changing if it lives up to its promise. The work is technically solid, distributed under an open-source licence, and open to contributions. The work is well presented, with a number of mostly-convincing use cases/demonstrations (but see comments

above on single neuron fitting results).

Suggested improvements

I would suggest the following improvements:

1. A discussion of previous work on spiking neuron simulators built on top of machine learning frameworks (Norse, BindsNET, snnTorch, etc.), and of how the approach used in JAXLEY relates to/goes beyond those tools.

Thank you for suggesting this! We have added such a discussion to the updated manuscript (new Methods section "Comparison to other toolboxes").

2. More discussion of the limitations/challenges of gradient descent in neural modelling, for example:

We have revised the manuscript by highlighting the limitations and opportunities for improvement you stated below. We also describe methods that could further improve gradient-based fitting.

We would like to highlight that we built JAXLEY to be as modular as possible. As such, JAXLEY is agnostic to the fitting procedure: JAXLEY provides the simulator and the ability to compute the gradient. The gradient can be used with any optimization method, ranging from gradient descent to advanced methods such as reinforcement learning, gradient-based Bayesian inference schemes, Kalman filtering, Pareto-front optimization, or simulated-annealing methods. We hope that JAXLEY will allow the community to develop, evaluate and improve fitting methods for biophysical models. We clarify this in the updated manuscript.

(i) The findings by Prinz and Marder that disparate mechanisms can give rise to very similar network activity (e.g., Prinz et al., 2004, which the manuscript cites but does not discuss), which challenge the idea of a "best fit".

Indeed, a vanilla application of JAXLEY, training with mean-squared error, will give just a single fit in these models, which does not capture biological diversity or degeneracy. JAXLEY can, however, be used to recover the full space of data-compatible models: As we demonstrate in Fig. 2h, Hamiltonian Monte-Carlo can recover the posterior distribution of parameters, thereby uncovering the full space of data-compatible parameters. The use of Hamiltonian Monte-Carlo (which enables Bayesian inference for models with many parameters) is enabled by the ability of JAXLEY to compute gradients. Similarly, JAXLEY can also be used to explore multiple solutions by simply initializing it at multiple different parameter values, and using gradient descent to find nearby local minima, or by exploiting parallelization for brute-force parameter scans.

(ii) The finding that the parameter spaces of complex neuron models are highly non-smooth, with narrow "valleys" or wormholes in parameter space connecting widely separated regions of good fit (e.g., Achard & De Schutter, 2006).

10/12

(iii) The concept of multi-objective optimisation, i.e. the observation that when fitting to multiple loss functions there may be a family of good models following a "Pareto front" (e.g., Druckmann et al., 2007; Jedlicka et al., 2022).

We have added such a discussion to the manuscript. Plain gradient descent will, indeed, not capture the Pareto-front of models, but we believe that JAXLEY will enable studies of how multi-objective optimization methods can be improved with gradient descent.

(iv) The common problem of getting stuck in local minima, and the associated need to add noise, for example through simulated annealing. It would be interesting to know if JAXLEY already provides functionality for this, or if this is a potential extension.

We agree that the loss landscape of biophysical models can be non-smooth, posing difficulties for any optimizer (including gradient descent). We have added a discussion regarding limitations and opportunities for fitting methods to the updated manuscript.

Nonetheless, our results demonstrate that even 'plain' gradient descent can perform well, and we developed a range of methods to deal with non-smooth loss surfaces:

We use curriculum learning to improve fitting for the recurrent neural network task (Fig. 4) and Polyak Gradient Descent on several tasks (e.g., Fig. 2a,d, Fig. 4) to improve robustness. In addition, as our improved results on single-cell fits show, GPU-parallelization across many gradient descent runs can further improve fitting quality.

I would also like more information on the "specific optimizers for non-convex loss surfaces", which it seems to me would be important for many problems in computational neuroscience. By "specific optimizers for non-convex loss surfaces", we referred to our modification to Polyak gradient descent, which is described in Methods. We have clarified this in the updated manuscript. As we describe in the Methods, we found that, due to the non-convexity of the loss-surface, the norm of the gradient could largely differ between steps (sometimes by orders of magnitudes). Polyak gradient descent overcomes this issue by dividing the gradient by its

gradient magnitude (to the power of a hyperparameter beta), thereby being less prone to variability in the gradient norm.

3. If this can be done in a reasonable amount of time, produce better fits for Figs 2d and S3 that reproduce the qualitative features seen in the experimental traces.

As described above, we revised the single neuron fits with increased computational budget and GPU acceleration.

4. Fix the upper/lower-case problems in the reference list.

We have fixed all references. Thank you for the pointer!

References

- Pehle Christian and Pedersen Jens Egholm (2021). Norse - A deep learning library for spiking neural networks, Zenodo, <https://doi.org/10.5281/zenodo.4422025>
- Hazan Hananel, Saunders Daniel J., Khan Hassaan, Patel Devdhar, Sanghavi Darpan T., Siegelmann Hava T. and Kozma Robert (2018), BindsNET: A Machine Learning-Oriented

11/12

Spiking Neural Networks Library in Python, *Frontiers in Neuroinformatics* 12:89
<https://doi.org/10.3389/fninf.2018.00089>

- Eshraghian Jason K, Ward Max, Neftci Emre, Wang Xinxin, Lenz Gregor, Dwivedi Girish, Bennamoun Mohammed, Jeong Doo Seok and Lu Wei D. (2023) Training spiking neural networks using lessons from deep learning. *Proceedings of the IEEE* 111:1016--1054

- Oláh VJ, Pedersen NP and Rowan MJM (2022) Ultrafast simulation of large-scale neocortical microcircuitry with biophysically realistic neurons *eLife* 11:e79535.

<https://doi.org/10.7554/eLife.79535>

- Wang T, Wang Y, Shen J, Wang L and Cao L (2022) Predicting spike features of Hodgkin-Huxley-type neurons with simple artificial neural network *Frontiers in Computational Neuroscience* 15:800875.

<https://doi.org/10.3389/fncom.2021.800875>

- Achard P, De Schutter E (2006) Complex Parameter Landscape for a Complex Neuron Model. *PLoS Comput Biol* 2(7): e94. <https://doi.org/10.1371/journal.pcbi.0020094>

- Druckmann S, Banitt Y, Gidon A, Schürmann F, Markram H, Segev I. A novel multiple objective optimization framework for constraining conductance-based neuron models by experimental data. *Front Neurosci*. 2007 Oct 15;1(1):7-18. doi:

10.3389/neuro.01.1.1.001.2007.

- Jedlicka Peter, Bird Alexander D. and Cuntz Hermann (2022). Pareto optimality, economy-effectiveness trade-offs and ion channel degeneracy: improving population modelling for single neurons. *Open Biol*.12220073

<http://doi.org/10.1098/rsob.220073>

- Prinz, A., Bucher, D. & Marder, E. Similar network activity from disparate circuit parameters. *Nat Neurosci* 7, 1345–1352 (2004). <https://doi.org/10.1038/nn1352>

- Review by Andrew P. Davison (<http://orcid.org/0000-0002-4793-7541>)

Reviewer #2 (Remarks on code availability):

I have successfully installed the JAXLEY package, but due to time limitations have not been able to thoroughly test it, or to inspect the code base in detail. With these provisos, the implementation appears to be very solid, actively developed, and following good practices such as the use of continuous integration testing.

Thank you for the positive feedback!

12/12

Version 1:

Decision Letter:

Our ref: NMETH-A57626A

30th Apr 2025

Dear Dr. Deistler,

Thank you for submitting your revised manuscript "JAXLEY: Differentiable simulation enables large-scale training of detailed biophysical models of neural dynamics" (NMETH-A57626A). It has now been seen by the original referees and their comments are below. The reviewers find that the paper has improved in revision, and therefore we'll be happy in principle to publish it in Nature Methods, pending minor revisions to satisfy the referees' final requests and to comply with our editorial and formatting guidelines.

TRANSPARENT PEER REVIEW

ORCID

Best regards,
Nina

Nina Vogt, PhD
Senior Editor
Nature Methods

Reviewer #1 (Remarks to the Author):

I already reviewed an earlier version of this manuscript in detail, and was very positive about it. With the additions to both Jaxley and the new manuscript, I am even more positive about the author's work. I thank them for addressing any concerns and look forward to seeing the manuscript published, so if not clear already, I endorse publication of this manuscript.

Reviewer #1 (Remarks on code availability):

I downloaded and tested the code, and read through major sections of it as well.

Reviewer #2 (Remarks to the Author):

The revised manuscript fully addresses my concerns and questions about the previous version. I would like to thank the authors for the time invested in responding to my review.

In summary, this is a very exciting advance for biologically detailed modelling and simulation in neuroscience, potentially game-changing if it lives up to its promise. The work is technically solid, distributed under an open-source licence, and open to contributions. The work is well presented, with a number of convincing use cases and demonstrations.

Review by Andrew P. Davison (<http://orcid.org/0000-0002-4793-7541>)

Dear Dr. Vogt,

We thank you and the reviewers for the constructive and detailed comments and for appreciating the importance of our work. We are excited about the assessment of our work as a “major breakthrough” (R1), “highly significant” (R2), that it “has potential to transform the practice of computational neuroscience, opening up avenues for investigation that could not be explored previously” (R2), and that one reviewer explicitly recommends publication (R1). We are particularly glad that the reviewers recognize our efforts in developing and maintaining JAXLEY as a user-friendly software, calling the software “clear, easy to read, and [...] straightforward for the community to use” (R1), as well as “very solid, actively developed, and following good practices” (R2).

Based on the comments of the reviewers, we performed several new analyses and included multiple improved explanations. In particular, we now

- provide an overview with differences to other simulators (R1 & R2),
- show improved single-neuron fits to experimental data (R2, see new Supp. Fig. S5),
- benchmark JAXLEY on additional compute platforms (R1, see new Supp. Fig. S3),
- demonstrate that JAXLEY can simulate a network of 12k neurons with simplified morphologies and 25M synapses on a single GPU (R1, see new Supp. Fig. S4).

In addition, based on the recommendations of the reviewers, we have extended the JAXLEY toolbox by adding the two additional features (including tutorials on how to use these features):

- support for ion diffusion (R1),
- support for simulating simplified neuron models (e.g., LIF, Izhikevich; R1).

We hope that our new results and modifications further strengthen the manuscript and are grateful for the reviewers’ constructive suggestions.

Sincerely, and on behalf of all authors

Michael Deistler
University of Tübingen

Jakob Macke
University of Tübingen

Reviewer #1 (Remarks to the Author):

As a computational neuroscientist for over 20 years, I am confident that the methods developed here are a major breakthrough both in terms of simulating biophysically detailed models of neurons and neuronal circuits, and as a way to optimize the models both to accurately replicate experimental data, and further to train detailed neuronal models to perform different types of behavioral tasks. As such, I would recommend having this manuscript published, with only minor modifications, and additional suggested tests below.

Thank you for the positive feedback! We hope that JAXLEY opens new possibilities to build biophysical models and to tune them to fit experimental data or solve computational tasks.

Although impressive, including more difficult tasks would highlight the strengths of the authors' approach. I would recommend the authors test their optimization algorithms for neuronal network circuit models, on other tasks similar to the following where models need to perform more complex behaviors in environments with dynamically changing variables, such as video game play, or navigation, etc. as illustrated in other recent papers: <https://doi.org/10.3389/fncom.2022.1017284>, <https://doi.org/10.1371/journal.pone.0265808>

Indeed, using gradient descent to optimize models on complex behavioral tasks is a major contribution of JAXLEY. Optimizing biophysical models on tasks as the one shown in these papers should be possible in principle. However, investigating highly challenging reinforcement learning tasks, like the ones proposed in these papers, and in particular in biophysically detailed models, would be beyond the scope of this paper, and likely requires full, separate studies. In general, however, JAXLEY can efficiently be combined with arbitrary learning paradigms, which we now highlight in the updated manuscript.

Another point of discussion is whether the methods illustrate anything about how real neuronal circuits learn or have been optimized through evolution. This would be important to highlight given that the authors found many of the parameter values/ranges did not shift substantially on average, between trained and untrained models, so perhaps the difference between functional and dysfunctional circuits is only marginal.

We do believe that Jaxley will enable neuroscientists to build more realistic mechanistic models which will shed light on learning in biological circuits. We note, however, that the gradient descent algorithms that JAXLEY uses to optimize models are not biologically plausible. Several recent studies have aimed to find and evaluate biologically plausible variants of, or alternatives to, backpropagation [1,2,3]. All of these approaches focused on networks with very simple neuron models. JAXLEY opens up opportunities for investigating these questions also in mechanistically realistic circuits. We have added the sentence "The ability to perform gradient descent may also provide insights into how biological systems learn or are optimized." to discuss this.

A second approach to studying this question, and as suggested by the reviewer, would be to inspect how the learning process shifts parameter values in mechanistic networks. While our simulations were not specifically set up to target this question, we nevertheless note that they show that—even with the same learning rule—different tasks lead to different results in this respect:

For the MNIST task, the distribution of parameters does indeed not change much (Fig. 5e), but individual parameters can change largely (but trained parameters remain correlated to

their initial values, matching findings in artificial neural networks [4]). This further strengthens our claim that the aggregate statistics of connectivity parameters in networks may not be sufficient to generate functional activity. We include this analysis in Supp. Fig. S11 of the updated manuscript:

In contrast, we also show that, on other tasks, the distribution of parameters can change strongly: for example, on the evidence-integration task with RNNs (Fig. 4f).

Therefore, additional clarification on the parameter distributions, whether particular neurons are more prominent in solving a specific task, would be important to understand how the modeled circuits learn to perform behaviors appropriately.

We agree that understanding how trained networks solve tasks will be an important area of research. We have begun to investigate such questions in Fig. 5g in the manuscript, where we inspected the tuning of hidden neurons in a task-trained network and found that some neurons code for individual digits after training.

The software in the github repository is clear, easy to read, and provides a large amount of documentation, and tutorials making it straightforward for the community to use the valuable tools the authors developed.

Thank you for appreciating our efforts in making JAXLEY a usable toolkit. We are committed to maintaining it and to making it the go-to toolbox for biophysical neuron modelling.

It will be useful to benchmark simulations on different hardware architectures, since this will motivate people to try the new software, depending on their hardware capabilities.

We have added benchmarks on two new hardware architectures: A MacBook Pro M3 CPU and a NVIDIA H100 GPU. The MacBook Pro was almost twice as fast as the previously used AMD CPU. The NVIDIA H100 had lower simulation time than an NVIDIA A100 and was twice as fast for very large batches of parameter sets processed in parallel. We have added the following figure to the updated manuscript, showing these results:

I would like some clarification/discussion on how the simulation software compares against other approaches/libraries for simulating neuronal circuits such as BINDSnet (<https://doi.org/10.3389/fninf.2018.00089>) and NetPyNE (<https://doi.org/10.7554/eLife.44494>) ?

Thank you for suggesting this! We have added a discussion to the updated manuscript (new Methods section “Comparison to other toolboxes”).

Some details on: Why are Izhikevich and integrate and fire neurons not included in the software library? These simplified (compared to HH) neurons are widely used. I would suggest including basic implementations of at least adaptive integrate-and-fire neurons in future versions of Jaxley.

We thank the reviewer for raising this. Indeed, in the original manuscript, Izhikevich and Integrate-and-fire neurons were not yet implemented as part of JAXLEY (due to its focus on detailed biophysical models). We have now added an implementation for (Leaky) integrate-and-fire neurons and for Izhikevich neurons (and we have clarified the statement on the documentation website). At the moment, JAXLEY does not yet implement and test surrogate gradient methods that can efficiently optimize these models (because of their spike discontinuity, direct gradients are not applicable), but we are planning to add this feature in the future.

Are synaptic weights in a fairly large circuit model modifiable with the GD optimization? What about optimization of architectural changes (number of neurons, number of layers) ? Can optimize synaptic "noise" inputs, which are often required to maintain activity in a circuit model?

JAXLEY can optimize synaptic weights in (very) large circuit models. As we show in Fig. 1f, JAXLEY can differentiate a circuit with 2k morphologically detailed neurons and 1M synapses, and in Fig. 5 we show that JAXLEY can optimize 50k synapses to fit MNIST. Following your question below, in a new Supp. Fig. S4, we now also show that JAXLEY can simulate and differentiate a circuit of 12k neurons with simplified morphologies (with Hodgkin-Huxley dynamics) and connected by 25M synapses, on a single GPU.

JAXLEY can also optimize synaptic “noise” inputs. In Fig. 5f, we show that JAXLEY can optimize input currents adversarially to “fool” the network into wrong classifications. Other optimization paradigms regarding (synaptic) inputs are also possible (e.g., to find maximally excitatory images [1]).

Architectural changes, however, cannot easily be optimized with gradient descent, as these are typically discrete and thus non-differentiable parameters (e.g., number of hidden neurons).

As such, optimizing architectures requires different optimizers such as evolutionary algorithms. JAXLEY supports arbitrary learning or update rules and can be used to implement such algorithms for optimizing architecture.

Does Jaxley support standard learning rules like hebbian plasticity, STDP, STDP/RL ? Those are important and widely used and it would be interesting to compare whether those rules can help with any of the proposed tasks.

Yes, JAXLEY flexibly supports diverse update rules. In building JAXLEY, we have ensured that it is a modular framework: It can simulate networks and compute their gradients with respect to inputs or parameters. How (or if) these gradients are used, however, is fully up to the user. Gradients can be plugged into a reinforcement learning algorithm, or they could be fully ignored in, e.g., a Hebbian learning paradigm. We believe that this flexibility of JAXLEY is one of its major strengths and will enable the community to develop and evaluate new powerful training methods for neural circuits. We clarify these abilities of JAXLEY in the manuscript.

Can Jaxley optimize parameters of intracellular dynamics, like calcium signaling (molecular interactions and reaction-diffusion) ? Is that supported or will it be supported in the future? Those types of dynamics are generally considered important to neuronal cellular function, but are often excluded from multiscale neural models that perform behaviors. Partly, that's due to difficulty in optimizing and simulating such models. Jaxley could provide a solution. I would suggest the authors could comment on these questions.

Indeed, JAXLEY can model and optimize parameters of intracellular dynamics. In Fig. 2a-e, we optimized the decay time constant of the calcium buffer and the fraction of free calcium, and any other factor of intracellular dynamics could, in principle, also be optimized.

Yet, at the time of submission of this manuscript, JAXLEY had not yet supported reaction-diffusion dynamics. To make even more detailed simulations possible, we have now implemented ion diffusion. The diffusion strength and ion dynamics can be optimized with gradient descent. We have written a tutorial on using these features on our documentation website (https://jaxley.readthedocs.io/en/latest/tutorials/11_ion_dynamics.html) and also mention this possibility in the manuscript.

How difficult would it be to optimize the type of model presented here (<https://doi.org/10.1016/j.celrep.2023.113378>) in Jaxley to replicate firing rates and physiological dynamics?

JAXLEY can optimize circuits of that scale on a single GPU. As a proof-of-principle demonstration, we have now set up a network of 12k neurons with three compartments each and connected by 25 million synapses. On one NVIDIA A100 GPU, running 100 ms of simulation took around 15 seconds, and computing the gradient with respect to all cellular conductances and all 25 million synaptic weights took 130 seconds. We have included this analysis as Supplementary Fig. S4 in the updated manuscript:

These types of models are now routinely constructed, but were written in other simulators (NEURON). So, can the NEURON modelers hope to benefit from Jaxley in the near term?

We agree that lowering barriers for usage is important for adoption of JAXLEY. We have already built a prototype for an automated tool which converts ion channel models from .nmodl (used by NEURON) to Python (such that they can be read by JAXLEY) and will release this tool soon. We are in discussions with the Allen Institute to provide support for using JAXLEY with the SONATA file format they use. We are in contact with several labs which are switching from NEURON to JAXLEY in order to find out how to support such transitions most effectively, and we are organising a tutorial on the upcoming CNS conference to reach more potential users. Overall, we hope that the computational power of JAXLEY, combined with our efforts to easily port and construct models, will allow NEURON users to easily transition to JAXLEY.

Reviewer #1 (Remarks on code availability):

Clear code with documentation, tutorials, example models provided in a github repository

References:

- [1] Ahmad, N., van Gerven, M. A., & Ambrogioni, L. (2020). Gait-prop: A biologically plausible learning rule derived from backpropagation of error. *Advances in Neural Information Processing Systems*, 33, 10913-10923.
- [2] Bengio, Y., Lee, D. H., Bornschein, J., Mesnard, T., & Lin, Z. (2015). Towards biologically plausible deep learning. *arXiv preprint arXiv:1502.04156*.
- [3] Bartunov, S., Santoro, A., Richards, B., Marris, L., Hinton, G. E., & Lillicrap, T. (2018). Assessing the scalability of biologically-motivated deep learning algorithms and architectures. *Advances in neural information processing systems*, 31.
- [4] Chizat, L., Oyallon, E., & Bach, F. (2019). On lazy training in differentiable programming. *Advances in neural information processing systems*, 32
- [5] Walker, Edgar Y., et al. "Inception loops discover what excites neurons most using deep predictive models." *Nature neuroscience* 22.12 (2019): 2060-2065.

Reviewer #2 (Remarks to the Author):

Summary of the key results

This manuscript presents JAXLEY, a novel simulator for detailed biophysical neuron models and networks thereof, built upon the open-source JAX library, developed by Google for high-performance numerical computing and machine learning.

By leveraging the capabilities of JAX for automatic differentiation, just-in-time compilation, and ease of parallelization using GPUs and TPUs (Tensor Processing Units), JAXLEY opens up the possibility of performing very rapid gradient-based optimisation across very large numbers of parameters, with applications to fitting models to data, and training networks of biologically-detailed neurons to perform useful tasks.

JAXLEY provides a Python-based programming environment that will be familiar to users of other biophysical neuron simulation tools, notably NEURON.

Originality and significance

Many other authors have built neural simulators on top of machine learning frameworks (PyTorch or Tensorflow), for example:

- Norse (<https://norse.github.io/norse/>; Pehle & Pederson, 2021)
- BindsNET (<https://bindsnet-docs.readthedocs.io/>; Hazan et al., 2018)
- snnTorch (<https://snntorch.readthedocs.io/>; Eshraghian et al., 2023)
- NengoDL/KerasSpiking (<https://www.nengo.ai/nengo-dl/>; <https://www.nengo.ai/keras-spiking/>)

As far as I am aware, all of these tools support mainly, or only, point-neuron models (e.g., integrate-and-fire (I&F) family, Izhikevich), and are focused on implementing traditional deep learning approaches using spiking neural networks.

Other researchers have targeted biophysical neural models, but not through directly solving the models' differential equations, rather by using recurrent neural networks to generate/predict the future activity of the neuron/network (Oláh et al., 2022; Wang et al., 2022).

Given this context of previous work, the principal points of originality of the current work appear to me as follows:

1. application of automatic gradient computation to biophysical, multi-compartment models, rather than to point neuron, I&F-type models;
2. a focus on fitting models to experimental data, alongside the more traditional deep learning capability of training networks to perform tasks;
3. possibly, implementation of the implicit Euler method in a machine learning framework (I have not investigated which integration methods are implemented by the tools cited above);
4. implementation on top of JAX, which apparently has generally better performance than PyTorch or Tensorflow.

While each of these points individually is a valuable advance, taken together they make the current study highly significant. While previous ML-based simulation tools have focused mainly on a machine learning audience, JAXLEY has the potential to transform the practice of computational neuroscience, opening up avenues for investigation that could not be explored previously due to lack of computational resources. This is not to discount its potential value in

machine learning approaches, studying for example the potential benefits of neuronal non-linearities in the computational capabilities and efficiency of deep and/or recurrent networks.

Thank you for appreciating the significance of our work! We would like to highlight the third point in your list of contributions: Implementing the implicit Euler method for morphologically detailed single neuron models requires efficiently solving a sparse linear system defined by the branching structure of a neuron (Hines, 1984). Implementing such a solver in a machine learning framework (and doing this in an efficiently parallelizable way, following, e.g., Ben-Shalom et al., 2022) is indeed a major contribution of our work.

Data & methodology: validity of approach, quality of presentation, clarity and context

I have successfully installed the JAXLEY package, but due to time limitations have not been able to thoroughly test it, or to inspect the code base in detail. With these provisos, the implementation appears to be very solid, actively developed, and following good practices such as the use of continuous integration testing.

Thank you for this positive feedback! We are committed to maintaining JAXLEY and to making it the go-to tool for biophysical neuron modelling.

While the results presented in this manuscript make me wish to try JAXLEY in a real world project, I am underwhelmed by the quality of the fitting results shown for single neurons. The fits in the left-panel of Figure 2d, and all of the panels in Supplementary Figure S3 are poor: the response of the fitted model does not capture the initial short burst, and the adaptation seen in the right-hand model in S3 is in the opposite direction to that seen in the experimental data (decreasing rather than increasing inter-spike intervals). This suggests either (i) a poor choice of loss function or (ii) the fitting has got stuck in a local minimum. I do not think the fits shown are a good advertisement for the capabilities of the software.

We agree that the fits shown in Fig. 2d and S3 are not optimal. In these fits, we had prioritized a proof-of-concept demonstration with low computational budget (roughly 2 minutes per cell on CPU).

To demonstrate that gradient descent can achieve closer fits, we have now made six additional parameters trainable, we modified the loss function, and we have scaled up fitting on a GPU. By running on GPU, we could run 1k gradient descent chains in parallel, allowing for a much better exploration of the parameter space. This procedure took 1-2h per neuron, and the resulting fits are shown below. We now achieve an excellent fit for three out of four cells, and a significantly improved fit for the last cell. Exploration of more complex fitting methods (e.g., by letting the gradient descent chains interact) or loss functions could further improve the fits.

Using JAXLEY, we also parallelized a genetic algorithm on GPU and found that – for these comparably simple models – it identified fits of similar quality. Finally, we note that these fits are significantly better than the best biophysical models provided on the Allen Cell-Types Database. However, the results are difficult to compare as the Allen-Cell-Types Database uses a different set of ion channels and optimizes its cells across a range of protocols. As such, we do not include the comparison to the Allen fits in the updated manuscript (but the rest of the figure is now available as Supplementary Fig. S6).

The writing and figures are generally clear and well presented, although I found the individual panels within the figures to be very small: I had to zoom in a lot to properly view them.

We used font size 6pt for all figures, which is within the recommendations of the Nature submission guide (5-7pt). Nonetheless, we agree that some panels in Fig. 2 had been relatively small, and we have adapted this figure (panels d,e,f,g).

There are some formatting problems in the references, with many cases of uppercase letters converted to lowercase (for example NEST->Nest, NEURON->neuron, BluePyOpt->Bluepyopt, GPU->gpu), suggesting use of BibTeX without appropriate escaping (by surrounding letters that should stay uppercase with curly braces {}).

Thank you for pointing this out. We have fixed all references.

Conclusions

In conclusion, this is a very exciting advance for biologically detailed modelling and simulation in neuroscience, potentially game-changing if it lives up to its promise. The work is technically solid, distributed under an open-source licence, and open to contributions. The work is well presented, with a number of mostly-convincing use cases/demonstrations (but see comments above on single neuron fitting results).

Suggested improvements

I would suggest the following improvements:

1. A discussion of previous work on spiking neuron simulators built on top of machine learning frameworks (Norse, BindsNET, snnTorch, etc.), and of how the approach used in JAXLEY relates to/goes beyond those tools.

Thank you for suggesting this! We have added such a discussion to the updated manuscript (new Methods section "Comparison to other toolboxes").

2. More discussion of the limitations/challenges of gradient descent in neural modelling, for example:

We have revised the manuscript by highlighting the limitations and opportunities for improvement you stated below. We also describe methods that could further improve gradient-based fitting.

We would like to highlight that we built JAXLEY to be as modular as possible. As such, JAXLEY is agnostic to the fitting procedure: JAXLEY provides the simulator and the ability to compute the gradient. The gradient can be used with any optimization method, ranging from gradient descent to advanced methods such as reinforcement learning, gradient-based Bayesian inference schemes, Kalman filtering, Pareto-front optimization, or simulated-annealing methods. We hope that JAXLEY will allow the community to develop, evaluate and improve fitting methods for biophysical models. We clarify this in the updated manuscript.

(i) The findings by Prinz and Marder that disparate mechanisms can give rise to very similar network activity (e.g., Prinz et al., 2004, which the manuscript cites but does not discuss), which challenge the idea of a "best fit".

Indeed, a vanilla application of JAXLEY, training with mean-squared error, will give just a single fit in these models, which does not capture biological diversity or degeneracy. JAXLEY can, however, be used to recover the full space of data-compatible models: As we demonstrate in Fig. 2h, Hamiltonian Monte-Carlo can recover the posterior distribution of parameters, thereby uncovering the full space of data-compatible parameters. The use of Hamiltonian Monte-Carlo (which enables Bayesian inference for models with many parameters) is enabled by the ability of JAXLEY to compute gradients. Similarly, JAXLEY can also be used to explore multiple solutions by simply initializing it at multiple different parameter values, and using gradient descent to find nearby local minima, or by exploiting parallelization for brute-force parameter scans.

(ii) The finding that the parameter spaces of complex neuron models are highly non-smooth, with narrow "valleys" or wormholes in parameter space connecting widely separated regions of good fit (e.g., Achard & De Schutter, 2006).

(iii) The concept of multi-objective optimisation, i.e. the observation that when fitting to multiple loss functions there may be a family of good models following a "Pareto front" (e.g., Druckmann et al., 2007; Jedlicka et al., 2022).

We have added such a discussion to the manuscript. Plain gradient descent will, indeed, not capture the Pareto-front of models, but we believe that JAXLEY will enable studies of how multi-objective optimization methods can be improved with gradient descent.

(iv) The common problem of getting stuck in local minima, and the associated need to add noise, for example through simulated annealing. It would be interesting to know if JAXLEY already provides functionality for this, or if this is a potential extension.

We agree that the loss landscape of biophysical models can be non-smooth, posing difficulties for any optimizer (including gradient descent). We have added a discussion regarding limitations and opportunities for fitting methods to the updated manuscript.

Nonetheless, our results demonstrate that even 'plain' gradient descent can perform well, and we developed a range of methods to deal with non-smooth loss surfaces:

We use curriculum learning to improve fitting for the recurrent neural network task (Fig. 4) and Polyak Gradient Descent on several tasks (e.g., Fig. 2a,d, Fig. 4) to improve robustness. In addition, as our improved results on single-cell fits show, GPU-parallelization across many gradient descent runs can further improve fitting quality.

I would also like more information on the "specific optimizers for non-convex loss surfaces", which it seems to me would be important for many problems in computational neuroscience.

By "specific optimizers for non-convex loss surfaces", we referred to our modification to Polyak gradient descent, which is described in Methods. We have clarified this in the updated manuscript. As we describe in the Methods, we found that, due to the non-convexity of the loss-surface, the norm of the gradient could largely differ between steps (sometimes by orders of magnitudes). Polyak gradient descent overcomes this issue by dividing the gradient by its gradient magnitude (to the power of a hyperparameter beta), thereby being less prone to variability in the gradient norm.

3. If this can be done in a reasonable amount of time, produce better fits for Figs 2d and S3 that reproduce the qualitative features seen in the experimental traces.

As described above, we revised the single neuron fits with increased computational budget and GPU acceleration.

4. Fix the upper/lower-case problems in the reference list.

We have fixed all references. Thank you for the pointer!

References

- Pehle Christian and Pedersen Jens Egholm (2021). Norse - A deep learning library for spiking neural networks, Zenodo, <https://doi.org/10.5281/zenodo.4422025>
- Hazan Hananel, Saunders Daniel J., Khan Hassaan, Patel Devdhar, Sanghavi Darpan T., Siegelmann Hava T. and Kozma Robert (2018), BindsNET: A Machine Learning-Oriented

Spiking Neural Networks Library in Python, *Frontiers in Neuroinformatics* 12:89

<https://doi.org/10.3389/fninf.2018.00089>

- Eshraghian Jason K, Ward Max, Neftci Emre, Wang Xinxin, Lenz Gregor, Dwivedi Girish, Bennamoun Mohammed, Jeong Doo Seok and Lu Wei D. (2023) Training spiking neural networks using lessons from deep learning. *Proceedings of the IEEE* 111:1016--1054

- Oláh VJ, Pedersen NP and Rowan MJM (2022) Ultrafast simulation of large-scale neocortical microcircuitry with biophysically realistic neurons *eLife* 11:e79535.

<https://doi.org/10.7554/eLife.79535>

- Wang T, Wang Y, Shen J, Wang L and Cao L (2022) Predicting spike features of hodgkin-huxley-type neurons with simple artificial neural network *Frontiers in Computational Neuroscience* 15:800875.

<https://doi.org/10.3389/fncom.2021.800875>

- Achard P, De Schutter E (2006) Complex Parameter Landscape for a Complex Neuron Model. *PLoS Comput Biol* 2(7): e94. <https://doi.org/10.1371/journal.pcbi.0020094>

- Druckmann S, Banitt Y, Gidon A, Schürmann F, Markram H, Segev I. A novel multiple objective optimization framework for constraining conductance-based neuron models by experimental data. *Front Neurosci.* 2007 Oct 15;1(1):7-18. doi: 10.3389/neuro.01.1.1.001.2007.

- Jedlicka Peter, Bird Alexander D. and Cuntz Hermann (2022). Pareto optimality, economy–effectiveness trade-offs and ion channel degeneracy: improving population modelling for single neurons. *Open Biol.*12220073

<http://doi.org/10.1098/rsob.220073>

- Prinz, A., Bucher, D. & Marder, E. Similar network activity from disparate circuit parameters. *Nat Neurosci* 7, 1345–1352 (2004). <https://doi.org/10.1038/nn1352>

- Review by Andrew P. Davison (<http://orcid.org/0000-0002-4793-7541>)

Reviewer #2 (Remarks on code availability):

I have successfully installed the JAXLEY package, but due to time limitations have not been able to thoroughly test it, or to inspect the code base in detail. With these provisos, the implementation appears to be very solid, actively developed, and following good practices such as the use of continuous integration testing.

Thank you for the positive feedback!